# Host-associated microbe PCR (hamPCR) enables convenient measurement of both microbial load and community composition

**Derek S Lundberg[1]\*, Pratchaya Pramoj Na Ayutthaya[1], Annett Strauß[2], Gautam Shirsekar[1], Wen-Sui Lo[3], Thomas Lahaye[3], Detlef Weigel[1]\***

[1]Department of Molecular Biology, Max Planck Institute for Developmental Biology, Tübingen, Germany; [2]Department of Evolutionary Biology, Max Planck Institute for Developmental Biology, Tübingen, Germany; [3]ZMBP-General Genetics, University of Tübingen, Tübingen, Germany

**Abstract** The ratio of microbial population size relative to the amount of host tissue, or 'microbial load', is a fundamental metric of colonization and infection, but it cannot be directly deduced from microbial amplicon data such as 16S rRNA gene counts. Because existing methods to determine load, such as serial dilution plating, quantitative PCR, and whole metagenome sequencing add substantial cost and/or experimental burden, they are only rarely paired with amplicon sequencing. We introduce host-associated microbe PCR (hamPCR), a robust strategy to both quantify microbial load and describe interkingdom microbial community composition in a single amplicon library. We demonstrate its accuracy across multiple study systems, including nematodes and major crops, and further present a cost-saving technique to reduce host overrepresentation in the library prior to sequencing. Because hamPCR provides an accessible experimental solution to the well-known limitations and statistical challenges of compositional data, it has far-reaching potential in culture-independent microbiology.

**\*For correspondence:**
derek.lundberg@gmail.com (DSL); weigel.elife@gmail.com (DW)

## Introduction

Knowing the relative abundance of individual taxa reveals important information about any ecological community, including microbial communities. An expedient means of learning their composition in a sample is to sequence and count a defined number of 16S or 18S rRNA genes (hereafter rDNA), the internal transcribed spacer (ITS) of rRNA arrays, or other amplicons that distinguish microbial species in a sample. However, these common amplicon counting-by-sequencing methods do not provide information on the density or load of the microbes. Critically, such microbial sequence counts lack a denominator accounting for the amount of the habitat sampled, and thus, sparsely-colonized and densely-colonized samples become indistinguishable, despite most study systems being open systems in which the total number of microbial cells can vary over many orders of magnitude. Another limitation of such compositional data is that because the sum of all microbes is constrained, an increase in the abundance of one microbe reduces the relative abundance of all other microbes, creating misleading interpretations in the absence of appropriate statistical methods (*Barlow et al., 2020*; *Gloor et al., 2017*; *Morton et al., 2019*; *Tsilimigras and Fodor, 2016*). Experimental determination of the microbial load, for example by relating microbial abundance to sample volume, mass, or surface area, has led to important insights in microbiome research that otherwise would have been missed with relative abundance data (*Humphrey and Whiteman, 2020*; *Niu et al., 2017*;

*Props et al., 2017*; *Regalado et al., 2020*; *Smets et al., 2016*; *Stämmler et al., 2016*; *Tkacz et al., 2018*; *Vandeputte et al., 2017*).

For many host-associated microbiome samples, in particular those from plants (*Regalado et al., 2020*), nematodes (*Ogier et al., 2020*), insects (*Ellegaard et al., 2020*; *Gendrin et al., 2015*; *Parker et al., 2020*), and other organisms in which it is difficult or impossible to physically separate microbes from host tissues, a thorough DNA extraction yields both host and microbial DNA. For such samples, the amount of DNA from host and microbe is directly proportional to the number of cells sampled (*Davies, 1977*; *Massonnet et al., 2011*), and therefore the ratio of microbial DNA to host DNA is an intrinsic measure of the microbial load of the sample (*Humphrey and Whiteman, 2020*; *Karasov et al., 2019*; *Karasov et al., 2018*; *Lebeis et al., 2015*; *Regalado et al., 2020*). Researchers have attempted to exploit this property and use the host rDNA amplified as a byproduct of microbial rDNA to calculate microbial load (*Humphrey and Whiteman, 2020*; *Lebeis et al., 2015*), but because host nuclear ribosomal arrays may have hundreds or thousands of copies (*Rabanal et al., 2017*), and organellar DNA is also overabundant, these methods are inefficient and require noisy interventions to increase the microbial signal. Sufficiently deep whole metagenome sequencing (WMS) also can in principle describe the microbial community composition and measure the microbial load, but is rarely practical because of a similar overrepresentation of host DNA (*Karasov et al., 2019*; *Regalado et al., 2020*). For example, WMS of a leaf extract from wild *Arabidopsis thaliana* typically yields >95% plant DNA and <5% microbial DNA. Furthermore, many WMS reads remain unclassifiable and thus unquantifiable in complex samples (*Karasov et al., 2019*; *Regalado et al., 2020*).

Most commonly, researchers combine amplicon sequencing with an additional orthogonal method. These include supplementary shallow WMS (*Regalado et al., 2020*), quantitative PCR (qPCR) or digital PCR of host and/or microbial genes (*Anderson and McDowell, 2015*; *Barlow et al., 2020*; *Ellegaard et al., 2020*; *Guo et al., 2020*; *Jian et al., 2020*; *Karasov et al., 2019*; *Nadkarni et al., 2002*), adding sequenceable 'spike-ins' calibrated based on sample volume (*Lin et al., 2019*), mass (*Stämmler et al., 2016*), or qPCR-determined host DNA content (*Guo et al., 2020*), counting colony-forming units (CFU)(*Chen et al., 2020*; *Niu et al., 2017*), and flow cytometry (*Props et al., 2017*; *Vandeputte et al., 2017*). The multitude of methods and publications hints at the enduring nature of this problem. While combining amplicon sequencing with any of these other approaches improves data, it requires more work, consumes more sample material, and introduces technical caveats, such as a reliance on accurately pipetting small quantities.

Here, we introduce host-associated microbe PCR or 'hamPCR', a robust and accurate single-reaction method to co-amplify a low-copy host gene and one or more microbial regions, such as 16S rDNA. We accomplish this with a two-step PCR protocol (*Carlson et al., 2013*; *Gohl et al., 2016*; *Lundberg et al., 2013*; *Symeonidi et al., 2020*; *Wen and Zhang, 2012*). In hamPCR, gene-specific primer pairs bind to the 'raw' templates in a first short step, which is run for only two cycles to limit propagating amplification biases related to primer annealing and primer availability. In the second exponential step, a single set of primers with complementarity to the universal overhangs add barcodes and sequencing adapters. Such co-amplification of diverse fragments is used in many RNA-seq and WMS protocols (*Kukurba and Montgomery, 2015*; *Quince et al., 2017*). Notably, *Carlson et al., 2013* similarly used a two-step PCR including a multiplexed first step of five to seven cycles to sequence and quantify both variable and joining segments at human T and B cell receptor loci, providing strong proof-of-concept for our method applied to the microbiome.

We designed our host and microbe amplicons to have slightly different lengths, such that they can be resolved by electrophoresis for quality control. We further show that after pooling finished sequencing libraries, the amplicons can be separately purified and re-mixed at any favorable ratio prior for sequencing (for example, with host DNA representing an affordable 5–10%), and sequence counts can be accurately scaled back to original levels in-silico. Thus, in stark contrast to WMS, samples with initially unfavorable host-to-microbe ratios can be easily adjusted *prior to sequencing* without loss of information. Because of the practical simplicity and flexibility of hamPCR, it has the potential to supplant traditional microbial amplicon sequencing in host-associated microbiomes.

## Results

### hamPCR generates quantitative sequencing-based microbial load

The first two-cycle host-and-microbe template tagging step ('HM-tagging') of hamPCR multiplexes two or more primer pairs in the same reaction, at least one of which targets a single- or low-copy host gene (*Appendix 1—figure 1*). The HM-tagging primers are then cleaned with Solid Phase Reversible Immobilization (SPRI) magnetic beads (*Rohland and Reich, 2012*; *Appendix 1—figure 2*). Next, an exponential PCR of 20–30 cycles is performed using universal barcoded primers (*Figure 1a*, *Appendix 1—figure 1*). As a host amplicon in *A. thaliana* samples, we targeted a

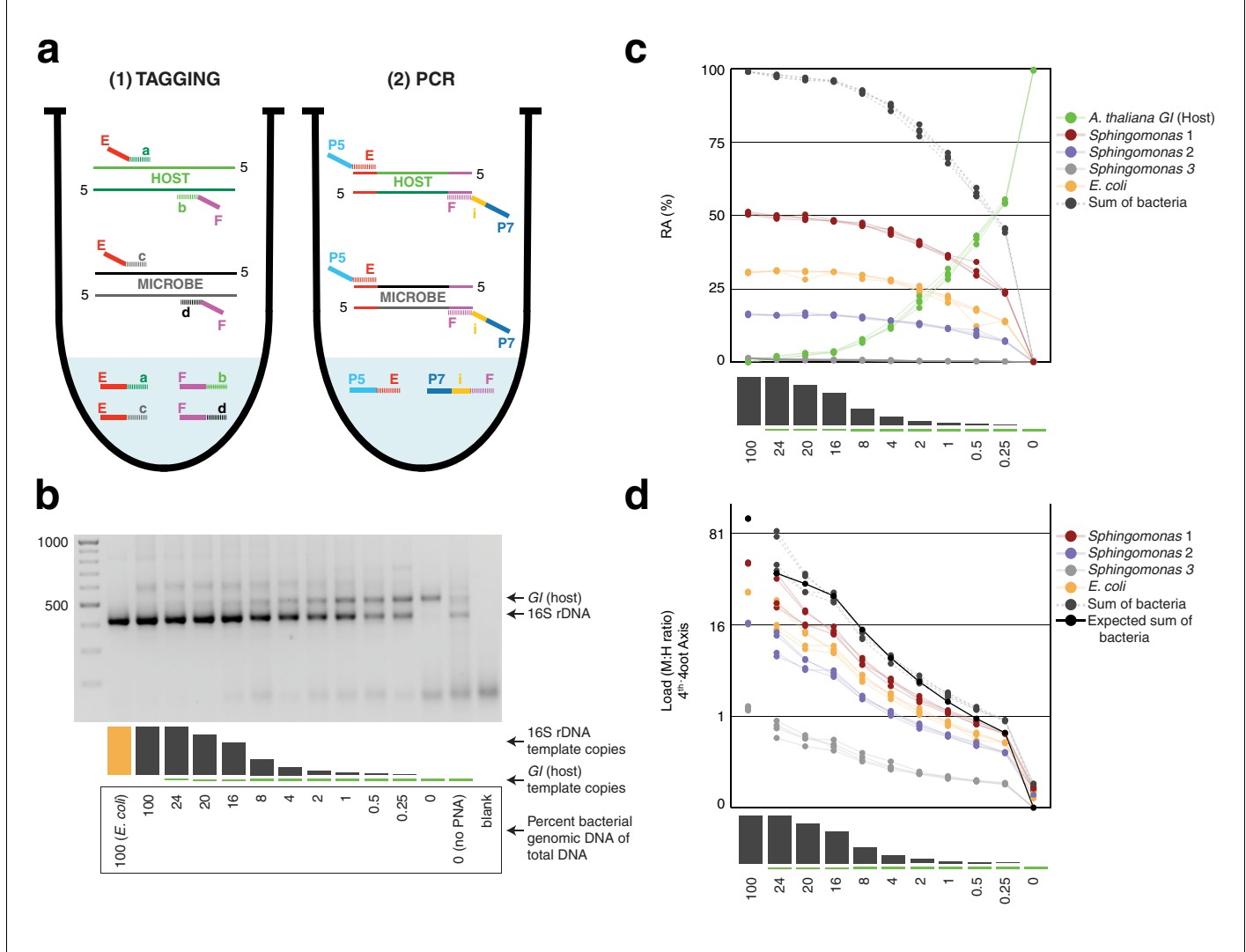

**Figure 1.** Synthetic samples demonstrate technical reproducibility. (a) Schematic showing the two steps of hamPCR. The HM-tagging reaction (left) shows two primer pairs: one for the host (E-a and F-b) and one for microbes (E-c and F-d). Each primer pair adds the same universal overhangs E and F. The PCR reaction (right) shows a single primer pair (P7-E and P5-i-F) that can amplify all tagged products. (b) Representative 2% agarose gel of hamPCR products from the synthetic titration panel, showing a V4 16S rDNA amplicon at ~420 bp and an *A. thaliana* GI amplicon at 502 bp. The barplot underneath shows the predicted number of original *GI* and 16S rDNA template copies. Numbers boxed below the barplot indicate the percent bacterial genomic DNA of total DNA. (c) Relative abundance of the host and microbial ASVs in the synthetic titration panel, as determined by amplicon counting. Pure *E. coli*, pure *A. thaliana* without PNAs, and blanks were excluded. (d) Data in (c) converted to microbial load by dividing by host abundance, with a fourth-root transformed y-axis to better visualize lower abundances.

The online version of this article includes the following figure supplement(s) for figure 1:

**Figure supplement 1.** Gel images from synthetic titration panel.

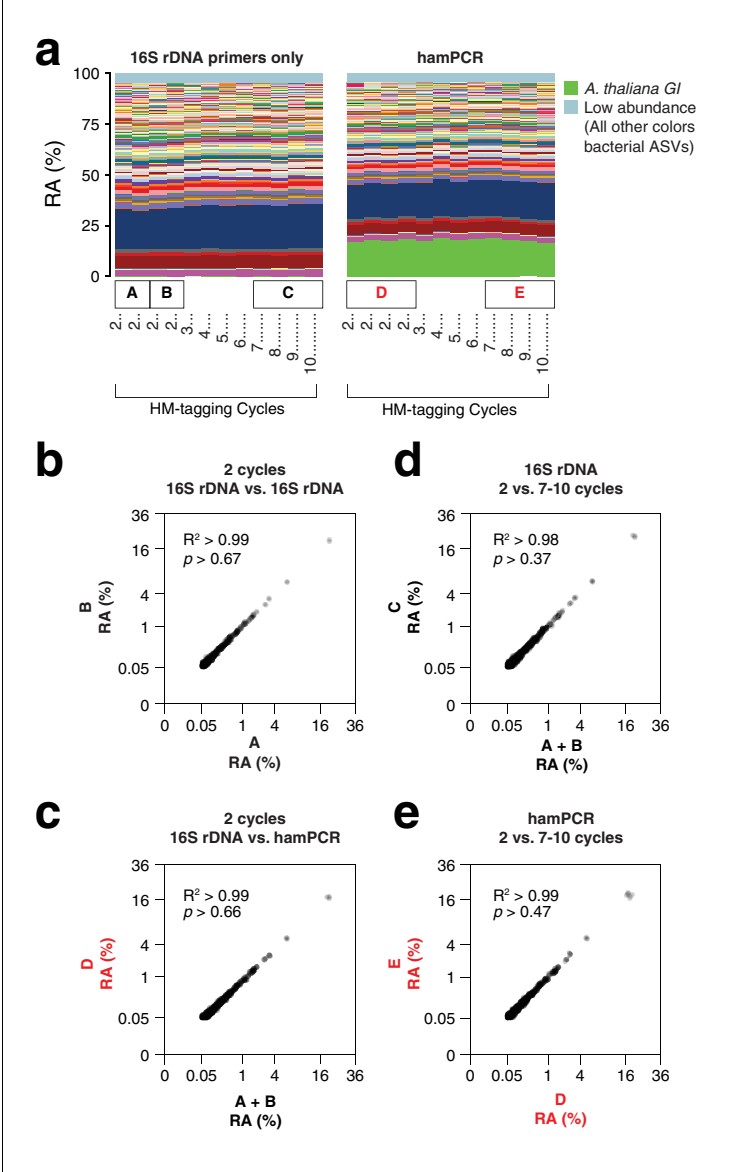

**Figure 2.** hamPCR is robust and does not distort a complex microbial community. (a) DNA extracted from wild *A. thaliana* phyllospheres was used as a template for both V4 16S rDNA PCR (left, 515F and 799R) and hamPCR (right, V4 16S rDNA and *GI* 502 bp primers). Four replicates were produced with two cycles of the HM-tagging reaction and 30 cycles of PCR, and additional replicates with 3 to 10 HM-tagging cycles paired with 29 to 22 PCR cycles (for a constant total of 32 cycles). The stacked columns show the relative abundances of major ASVs. Boxed upper case letters demarcate groups of samples compared below. (b) Correlation of fourth-root transformed ASV abundances for the 16S rDNA samples above panel (a) box [A] to the 16S rDNA samples above box [B]. Only ASVs with a minimum relative abundance of 0.05% were compared. $R^2$, coefficient of determination. *p*-value from Kolmogorov-Smirnov test. (c) Same as (b), but for the four 16S rDNA samples above box [A] and [B] compared to the four hamPCR samples above box [D]. For hamPCR, the *A. thaliana GI* ASV was removed and the bacterial ASVs were rescaled to 100% prior to the comparison. (d) Same as (b) and (c), but for the four 16S rDNA samples above box [A] and [B] compared to the 16S rDNA samples above box [C]. (e) Same as (b), (c), and (d), but for the four hamPCR samples above box [D] compared to the four hamPCR samples above box [E].

The online version of this article includes the following figure supplement(s) for figure 2:

**Figure supplement 1.** Gel of template HM-tagging cycling tests.

**Figure supplement 2.** Effect of varying HM-tagging cycles on the plasmid 1:1 host:microbe template.

fragment of the *GIGANTEA* (*GI*) gene, which is well conserved and present as a single copy in *A. thaliana* and many other plant species (*Duarte et al., 2010*). As microbial amplicons, we initially targeted widely used regions of 16S rDNA. Further considerations for primer design are discussed in Appendix 1.

To assess the technical reproducibility of the protocol, we made a titration panel of artificial samples combining varying amounts of pure *A. thaliana* plant DNA with pure bacterial DNA that reflects a simple synthetic community (Materials and methods). These represented a realistic range of bacterial concentrations as previously observed from WMS of wild leaves, ranging from about 0.25% to 24% bacterial DNA (*Regalado et al., 2020*). All DNA preps employed heavy bead beating to ensure thorough lysis of both host and microbes, as an incomplete DNA extraction can lead to underrepresentation of hard-to-lyse cells (*Albertsen et al., 2015*; *Yuan et al., 2012*). We applied hamPCR to the panel, pairing each of three commonly-used 16S rDNA amplicons for the V4, V3V4, and V5V6V7 variable regions with either a 502 bp or 466 bp *GI* amplicon (Materials and methods, *Supplementary file 1*), such that the host and microbial amplicons differed by approximately 80 bp in length and were resolvable by gel electrophoresis. In all pairings, the *GI* band intensity increased as the 16S rDNA band intensity decreased (*Figure 1b*, *Figure 1—figure supplement 1*).

Focusing on the V4 16S rDNA primer set, 515F - 799R, paired with the 502 bp *GI* amplicon, we amplified the entire titration panel in four independently-mixed technical replicates. Although plant organelle sequences can be removed bioinformatically, we attempted to block their amplification as much as possible. In addition to use of the chloroplast-avoiding 799R primer (*Chelius and Triplett, 2001*), plant organelle-blocking peptide nucleic acids (PNAs) (*Lundberg et al., 2013*) further prevented unwanted 16S rDNA signal from organelles in the pure plant sample (*Figure 1b*). We note that although these PNAs are widely used and extremely effective, they do not work for all plant hosts (*Fitzpatrick et al., 2018*) and they can interfere with analysis of certain bacteria present in some environments (*Jackrel et al., 2017*). We pooled the replicates and sequenced them as part of a paired-end HiSeq 3000 lane. Because the 150 bp forward and reverse reads were not long enough to assemble into full amplicons, we analyzed only the forward reads (Materias and methods), processing the sequences into Amplicon Sequence Variants (ASVs) and making a count table of individual ASVs using *Usearch* (*Edgar, 2010*).

After identifying the ASVs corresponding to host *GI* and the bacteria in the synthetic community, we plotted the relative abundance of *A. thaliana GI,* the three *Sphingomonas* ASVs, and the single *E. coli* ASV across the samples of the titration panel (*Figure 1c*). There was high consistency between the four replicates, more than what was visually apparent in the gel (*Figure 1c*, *Figure 1—figure supplement 1*). We next divided ASV abundances in each sample by the abundance of the host ASV in that sample to give the quantity of microbes per unit of host, a measure of the microbial load. Plotting the data with a fourth-root transformed Y axis for better visualization of low bacterial loads, we observed consistent and accurate quantification of absolute microbial abundance from 0 up to about 16% total bacterial DNA (*Figure 1d*). Through this range, the actual sequence counts for total bacteria matched theoretical expectations based on the volumes pipetted to make the titration (solid black line, *Figure 1d*). At 16% bacterial DNA, bacteria contributed more than 96% of sequences, and the microbe-to-host template ratio was near 25. At higher microbial loads the trend was still apparent, and the decrease in precision was likely exacerbated by the effects of small numbers; when the host ASV abundance is used as a denominator and the abundance approaches 0, load approaches infinity and sampling error has a greater and greater influence on the quotient. Eventually, this creates unacceptable uncertainty. We defined a 'noise factor' $N$ as the full range in microbial load quotients that would result from adding a single host count and subtracting a microbial count from a sample ([microbe counts + 1] / [host counts - 1]) or vice versa ([microbe counts - 1] / [host counts + 1]). $N$ increases as microbial load increases, but this is overcome with increased total sequencing depth. We determined conservatively that samples for which $N > 0.22$ should only be classified as 'highly colonized', and should not be used for quantitative measurements (*Appendix 1—figure 3*). In our case, only a minority of highly infected hosts reached bacterial abundances above the highly quantitative range.

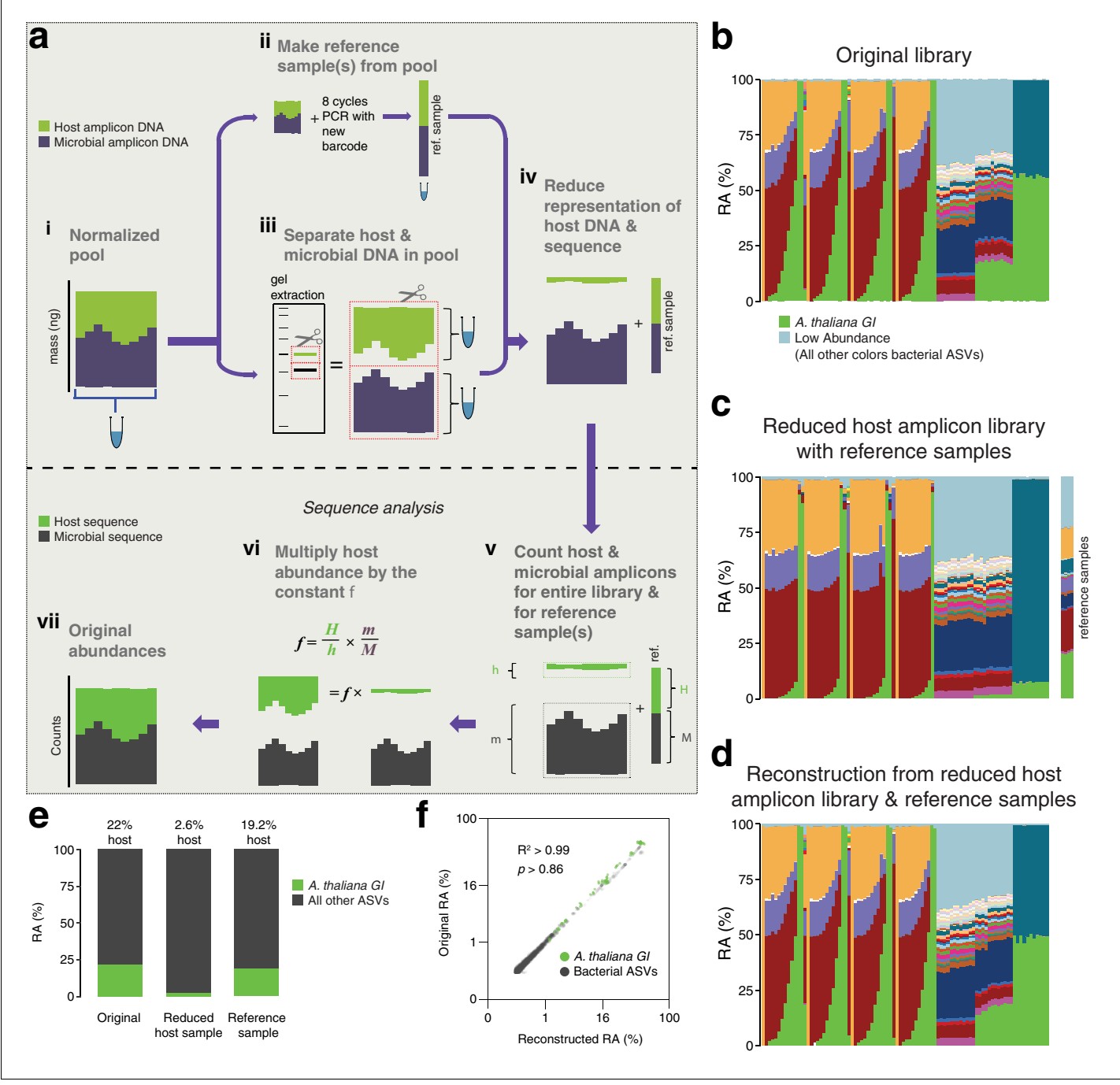

**Figure 3.** After remixing hamPCR amplicons for efficient sequencing, original abundances can be reconstructed. (**a**) Scheme of remixing process. (**i**): Products of individual PCRs are pooled at equimolar ratios into a single tube. (**ii**): An aliquot of DNA from the pool in (**i**) is re-amplified with eight cycles of PCR to replace all barcodes in the pool with a new barcode, creating a reference sample. (**iii**): An aliquot of the pool from (**i**) is physically separated into host and microbial fractions via agarose gel electrophoresis. (**iv**): The host and microbial fractions and the reference sample are pooled in the ratio desired for sequencing. (**v**): All sequences are quality filtered, demultiplexed, and taxonomically classified using the same parameters. (**vi**): Host and microbial amplicon counts are summed from the samples comprising the pooled library (*h* and *m*, respectively), and from the reference sample (*H and M*). (**vi**): *H, h, M, and m* are used to calculate the scaling constant *f* for the dataset. All host sequence counts are multiplied by *f* to reconstruct the original microbe-to-host ratios. (**vii**): Reconstructed original abundances. (**b**) Relative abundance (RA) of actual sequence counts from our original HiSeq 3000 run. (**c**) Relative abundance of actual sequence counts from our adjusted library showing reduced host and four reference samples. (**d**) The data from (**c**) after reconstructing original host abundance using the reference samples. (**e**) The total fraction of host vs. other ASVs in the original library, reduced host library, and reconstruction. (**f**) Relative abundances in the original and reconstructed library for all ASVs with a 0.05% minimum abundance, shown on fourth-root transformed axes. $R^2$, coefficient of determination. p-Value from Kolmogorov–Smirnov test.

*Figure 3 continued on next page*

*Figure 3 continued*

The online version of this article includes the following figure supplement(s) for figure 3:

**Figure supplement 1.** Library remixing prior to sequencing.
**Figure supplement 2.** Gels and Bioanalyzer traces showing steps of remixing.

## hamPCR does not distort the detected composition of the microbial community

We amplified products from a wild *A. thaliana* phyllosphere template DNA preparation (*Regalado et al., 2020*), either with four technical replicates using V4 16S rDNA primers alone, or alternatively with four technical replicates using hamPCR. After sequencing and deriving ASVs, we first compared ASV abundances within identically-prepared replicates of the pure 16S rDNA protocol to demonstrate best-case technical reproducibility of this established technique. As expected, this resulted in a nearly perfect correlation, with a coefficient of determination $R^2$ of 0.99 and abundance distributions that were indistinguishable by a Kolmogorov–Smirnov test (*Figure 2b*). Next, we removed the ASV corresponding to *A. thaliana GI* from the hamPCR data and rescaled the remaining microbial ASVs to 100% to give relative abundance data. We then compared microbial ASVs from the four pure 16S rDNA replicates to those from the four rescaled hamPCR replicates. In this comparison as well, $R^2$ was 0.99 and the distributions were essentially identical (*Figure 2c*). Thus, the inclusion of a host amplicon in the reaction did not introduce taxonomic biases.

## Sensitivity to number of HM-tagging cycles and template concentration

Two HM-tagging cycles minimize amplification biases that might otherwise have compounding effects due to differential primer efficiencies for the host and microbial templates. However, for templates at borderline low concentrations, inefficiencies due to SPRI cleanup could represent a bottleneck in amplification. Additionally, some techniques that prevent off-target organelle amplification (*Agler et al., 2016b*; *Song and Xie, 2020*) may benefit from additional HM-tagging cycles. To investigate the sensitivity of the results to additional HM-tagging cycles, we applied hamPCR for 2 through 10 HM-tagging cycles, both on the wild *A. thaliana* phyllosphere DNA described above and on a synthetic plasmid-borne template that contains bacterial rDNA and a partial *A. thaliana GI* gene template in cis in a 1:1 ratio (Appendix 1 - Discussion 2). Surprisingly, for the primers used here, there was no apparent influence of additional HM-tagging cycles, as 7–10 HM-tagging cycles yielded the same distribution of host and 16S rDNA ASV abundances as two cycles (Kolmogorov-Smirnov test, p > 0.47). This was true for hamPCR and for 16S rDNA primers alone (*Figure 2d and e*). This ideal result may not be achievable for all primer pairs and should be tested experimentally, but it is consistent with data that either 5 or 7 HM-tagging cycles gave comparable results for quantifying the human immune receptor repertoire (*Carlson et al., 2013*), and with the fact that properly designed multiplex reactions can be used in qPCR that is carried out with many cycles (*Vet et al., 1999*). We noticed that application of hamPCR to the 1:1 synthetic template yielded an average of 56.5% host *GI* and 43.5% bacteria, invariant with HM-tagging cycle number (*Figure 2—figure supplement 1* and *Figure 2—figure supplement 2*). This slight and consistent bias in favor of *GI* may be a result of slight differences in HM-tagging primer efficiency or primer concentration, and should be fine-tunable by altering primer concentration (*Carlson et al., 2013*; *Appendix 1—figure 4*).

As a further exploration of the robustness of the protocol, we applied hamPCR to a range of total *A. thaliana* leaf template concentrations of between 5 and 500 ng total DNA per reaction, covering a typical template range of 5–100 ng. Through the typical range, there was no difference in microbe or host ASV abundances. At 200 ng or above, the host amplicon seemed to be slightly favored, possibly because the 16S rDNA primers started to become limiting at these concentrations (*Appendix 1—figure 5*).

## Pre-sequencing adjustment of host-to-microbe ratio

Some host DNA must be present so that microbial load can be calculated, but sequencing too much host DNA would add unnecessary expense. We realized that the size difference between host and microbe bands in hamPCR affords not only independent visualization of both amplicons on a single

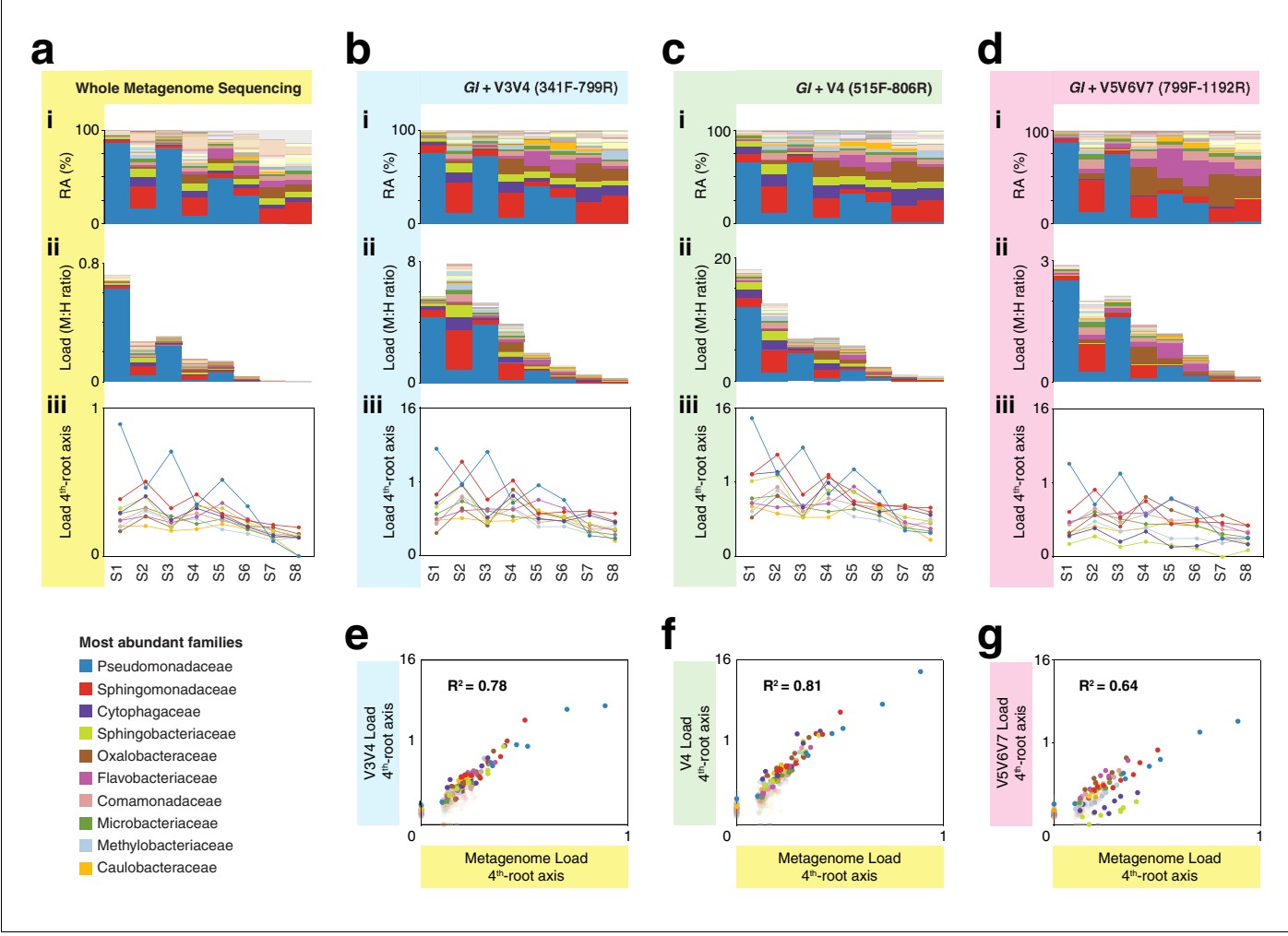

**Figure 4.** hamPCR with three common 16S rDNA amplicons gives consistent results that agree with whole metagenome sequencing (WMS). (**a**) (**i**): Stacked-column plot showing the relative abundance (RA) of bacterial families in eight wild *A. thaliana* leaf samples, as determined by WMS. The families corresponding to the first 10 colors from bottom to top are shown in reverse order on the bottom left. (**ii**) Stacked-column plot showing the bacterial load of the same bacterial families (M:H ratio = microbe-to-host ratio). (**iii**) The M:H bacterial load ratios for the 10 major bacterial families shown on a fourth-root transformed y-axis. Lines across the independent samples are provided as a help to visualize patterns. (**b**) Similar to (**a**), but with abundances resulting from hamPCR targeting a 502 bp *A. thaliana GI* amplicon and a ~590 bp V3V4 16S rDNA amplicon. (**c**) Similar to (**b**), but with the 16S rDNA primers targeting a ~420 bp V4 16S rDNA amplicon. (**d**) Similar to (**b**), but with a 466 bp *A. thaliana GI* amplicon and a ~540 bp V5V6V7 16S rDNA amplicon. (**e**) Fourth-root transformed abundance of each bacterial family determined by hamPCR of V3V4 16S rDNA plotted against the fourth-root transformed bacterial load from WMS. $R^2$ = Coefficient of determination. (**f**) Same as (**e**), but for hamPCR of V4 16S rDNA. (**g**) Same as (**e**), but for V5V6V7 16S rDNA.

The online version of this article includes the following figure supplement(s) for figure 4:

**Figure supplement 1.** Gel pictures from hamPCR applied to wild *A. thaliana* leaf DNA samples.

gel, but also allows convenient and easy adjustment of the host and microbial signals in the pooled library prior to sequencing, in order to improve cost effectiveness. We developed a strategy by which the final hamPCR amplicons are pooled and one aliquot of the pool is rebarcoded to form a reference sample that preserves the original host- to- microbe ratio. The remainder of the pool is run on a gel and the host and microbial bands are separately purified, quantified, and remixed (e.g. to reduce host and gain more microbial resolution). The rebarcoded reference sample, which was not remixed and thereby preserves the original ratio, can be sequenced separately or spiked into the remixed library prior to sequencing. Following sequencing, the reference sample provides the key to the correct host and microbe proportions, allowing simple scaling of the entire library back to original levels (*Figure 3a*, *Figure 3—figure supplement 1*).

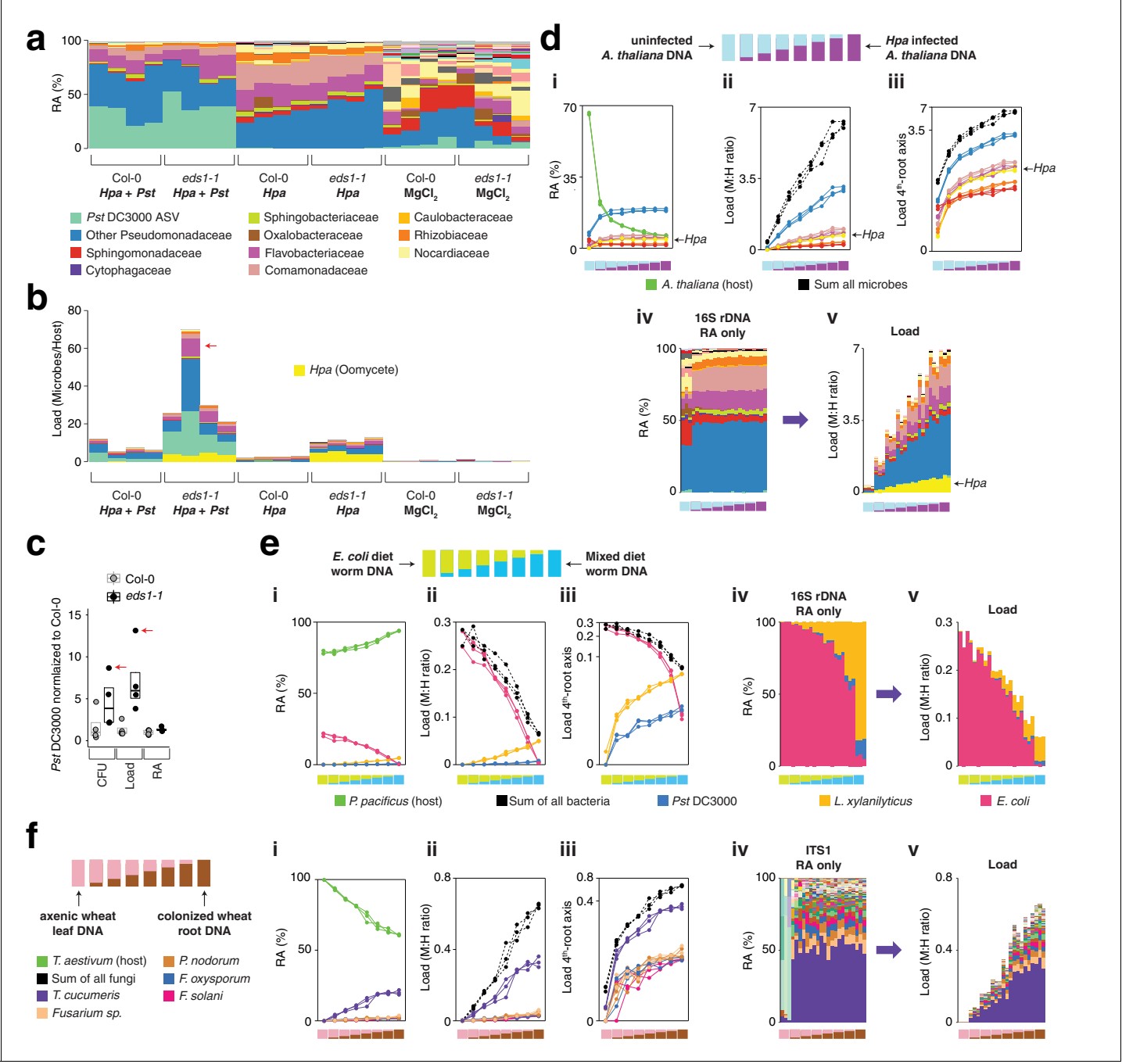

**Figure 5.** hamPCR can be generalized to more than two amplicons, non-plant hosts, and large or polyploid host genomes. (a) Relative abundance (RA) of only 16S rDNA amplicons for plants co-infected with *Hpa* and *Pst* DC3000 and their controls. Each column represents an independent plant. The ASV corresponding to *Pst* DC3000 (light green) is shown at the bottom and separately from other Pseudomonadaceae; all other bacteria are classified to the family level, including remaining Pseudomonadaceae. *Hpa* is not detectable using 16S rDNA primers and is therefore not shown. (b) The same data as shown in (a), but making full use of hamPCR by including the ASV for *Hpa* and converting the combined measurements to microbial load. The red arrow indicates an outlier sample. Same color key as in (a), with an additional color (yellow) for *Hpa* added. *Hpa* amplicon abundance was scaled by a factor of 4 in this panel for better visualization. (c) *Pst* DC3000 bacteria were quantified in parallel on the Col-0 and *eds1-1* samples infected with *Pst* DC3000 using CFU counts, the microbial load data in (b), or the relative abundance data in (a). The median is shown as a horizontal line and box boundaries show the lower and upper quartiles. Red arrows indicate the same outlier sample shown in (b). (d) An uninfected plant sample was titrated into an *Hpa*-infected sample to make a panel of eight samples. (i): the relative abundance of hamPCR amplicons with median abundance above 0.15%. (ii): after using host ASV to convert amplicons to load. The cumulative load is shown in black. (iii): the load on a fourth-root transformed y-axis, showing less-abundant families. (iv): stacked column visualization of all ASVs for the panel as it would be seen with pure 16S rDNA data. (v): stacked-column plot

*Figure 5 continued on next page*

Figure 5 continued

of the panel corrected for microbial load. Same color key as in (b), but with colors for *A. thaliana* and sum of microbes added. (e) Similar to (d), but with the nematode worm *P. pacificus* as host, and V5V6V7 16S rDNA primers. Instead of bacterial families, specific ASV abundances are shown. (f) Similar to (e), but with hexaploid wheat *T. aestivum* as host, and fungal ASV abundances from ITS1 amplicons.

The online version of this article includes the following figure supplement(s) for figure 5:

**Figure supplement 1.** Gel pictures of hamPCR applied to one host and two microbial amplicons.
**Figure supplement 2.** Gel of *P. pacificus* hamPCR titration libraries.
**Figure supplement 3.** Gels of *T. aestivum* hamPCR titration libraries.
**Figure supplement 4.** ITS1 load in *T. aestivum* with a 2:1 microbe-to-host HM-tagging primer ratio, and 16S rDNA load in *T. aestivum*.

To demonstrate this concept, we prepared four replicate reference samples for our HiSeq 3000 run, which included much of the data from *Figure 1* and *Figure 2*, and then separately purified the host and microbial fractions of the library (*Figure 3—figure supplements 1* and *2*). Based on estimated amplicon molarities of the host and microbial fractions, we remixed them, aiming for a sufficient but cost-efficient amount of 5% host DNA, added the reference samples, and sequenced the final mix as part of a new HiSeq 3000 lane. A stacked-column plot of relative abundances for all samples on the original run clearly showed the host *A. thaliana GI* ASV highly abundant in some samples, on average responsible for about 22% of total sequences in the run (*Figure 3b and e*). The remixed reduced host library had nearly 10-fold less total host *GI* ASV, 2.6%, slightly lower than our target of 5% (*Figure 3c and e*). The reference samples averaged 19.2% of host *GI* ASV, very close to the 22% host fraction in the original library. After using the reference samples to reconstruct the original host abundance in the remixed dataset, we recreated the shape of the stacked column plot from the original library (compare *Figure 3b–d*). When the fourth-root abundances for ASVs above a 0.05% threshold were compared between the original and reconstructed libraries, the $R^2$ coefficient of determination was 0.99, with no significant difference between the distributions (Kolmogorov-Smirnov test, p > 0.86). Thus, if hamPCR libraries have already been prepared and the host amplicon is overabundant, the host representation in the pooled libraries can be easily and accurately reduced using basic agarose gel technologies to enable more efficient sequencing.

## hamPCR with different 16S rDNA regions compared to whole metagenome sequencing

We next applied hamPCR to leaf DNA from eight wild *A. thaliana* plants that we had previously analyzed by WMS, and from which we therefore had an accurate estimate of the microbial load as the number of microbial reads divided by the number of plant chromosomal reads (*Regalado et al., 2020*). We applied hamPCR with primer combinations targeting the host *GI* gene and either the V3V4, V4, or V5V6V7 variable regions of the 16S rDNA. We produced three independent replicates for each primer set, which we averaged for final analysis (*Figure 4*, *Figure 4—figure supplement 1*). Across WMS and the three hamPCR amplicon combinations, the relative abundance of bacterial families was consistent (*Figure 4a–d*: i), with slight deviations likely due to the different taxonomic classification pipeline used for the metagenome reads (*Regalado et al., 2020*), as well as known biases resulting from amplification or classification of different 16S rDNA variable regions (*Graspeuntner et al., 2018*; *Thijs et al., 2017*). After converting both WMS and hamPCR bacterial reads to load by dividing by the plant read count in each sample, we recovered a similar pattern despite the quantification method, with decreasingly lower total loads progressing from plant S1 to plant S8, and individual bacterial family loads showing similar patterns (*Figure 4a–d*: ii, iii). Relative differences in load estimates when comparing the different hamPCR amplicons are likely in part due to different affinities of the 16S rDNA primer pairs for their targets in different bacterial species, and rDNA copy number variation among the microbial families (*Kembel et al., 2012*). To quantify the consistency of hamPCR load estimates with WGS load estimates, we plotted the loads against each other and found strong positive correlations, with hamPCR using V4 rDNA having the highest correlation to WGS (*Figure 4e–g*).

It is important to note that while relative load ratios between samples were consistent across hamPCR primer sets, the total microbe-to-host ratio varied substantially, with the maximum V5V6V7 16S rDNA total load at less than three times host, and the maximum V4 16S rDNA total load near 16

times host. This is likely due to variation in *GI* and 16S rDNA primer efficiencies. To make a statement about the ratio of plant cells to bacterial cells using hamPCR, it would be important to include standard samples with known bacterial load ratios, and to normalize each bacterial taxon by its average rDNA copy number.

## Three-amplicon hamPCR for simultaneous determination of oomycete and bacterial load

One disadvantage of 16S rDNA primers is that they readily amplify sequences from bacterial species, but their targets are absent from other microbes such as fungi, oomycetes, and archaea; other primer sets must therefore be used to detect these other groups. We attempted to overcome this limitation by setting up hamPCR not only with 16S rDNA and host *GI* sequences as described above, but also adding a third primer pair targeting oomycetes, which include important pathogens of *A. thaliana*. We first tested universal ITSo primers broadly targeting oomycete internal transcribed spacer rDNA (Materials and methods) in combination with *A. thaliana GI* primers and 16S rDNA primers targeting the bacterial V4, V3V4, or V5V6V7 regions, using as template our synthetic plasmid that includes templates for the three primer sets in equal proportion (Materials and methods, Appendix 1 - Discussion 2). A combination of all three amplicons seemed to work efficiently for the V4 region (*Figure 5—figure supplement 1*), and with this encouraging result, we set up a simple infection experiment. As pathogens, we prepared local strain 466–1 of the obligate biotrophic oomycete *Hyaloperonospora arabidopsidis* (*Hpa*) (*Coates and Beynon, 2010*) and the well-described bacterial pathogen *Pseudomonas syringae* pv. *tomato* (*Pst*) DC3000 (*Xin and He, 2013*). We used two *A. thaliana* genotypes: the reference accession Col-0, which is resistant to *Hpa* 466–1 but susceptible to *Pst* DC3000, and an *enhanced disease susceptibility 1* (*eds1-1*) mutant, which has a well-studied defect in a lipase-like protein necessary for many disease resistance responses and which is susceptible to both pathogens (*Bhandari et al., 2019*).

We infected seedlings with either *Hpa* 466–1 alone, a mix of *Hpa* 466–1 and *Pst* DC3000, or a buffer control, and maintained them for 7 days under cool, humid conditions ideal for *Hpa* growth (Materials and methods). The *eds1-1* plants inoculated with *Hpa* 466–1 became heavily infected and sporangiophores were too numerous to count. No visible bacterial disease symptoms were present on any of the plants, likely because the cool temperature decelerated bacterial growth and symptom appearance (*Huot et al., 2017*). We ground pools of four to five seedlings in a buffer and used a small aliquot to count *Pst* DC3000 CFUs, and the remainder of the lysate for DNA isolation and hamPCR. Despite the lack of bacterial symptoms, we recovered *Pst* DC3000 CFUs from the inoculated plants.

We applied hamPCR to these samples using the ITSo/16S/*GI* primer set, but due to excessive ITSo product, we repeated library construction replacing the ITSo primers with primers for a single copy *Hpa* actin gene (*Figure 5—figure supplement 1*; *Anderson and McDowell, 2015*). Intensity of the actin product correlated with visual *Hpa* symptoms (*Figure 5—figure supplement 1*). Sequencing the libraries confirmed *Hpa* and *Pst* ASVs in the inoculated samples, as expected. A standard bacterial relative abundance plot, as would be obtained from pure 16S rDNA data, confirmed the presence of *Pst* DC3000 in the bacteria-infected samples, and in addition revealed that *Hpa*-infected samples had a different bacterial community than uninfected samples (*Figure 5a*). Importantly, it failed to detect obvious differences between microbial communities on Col-0 and *eds1-1* plants. However, after including the actin ASV from *Hpa* and converting all abundances to microbial load, a striking difference became apparent between Col-0 and *eds1-1*, with *eds1-1* supporting higher bacterial and *Hpa* abundances (*Figure 5b*). This is expected from existing knowledge (*Bhandari et al., 2019*), and supported by *Pst* DC3000 CFU counts from the same plants (*Figure 5c*). The microbial load plot also revealed that *Hpa*-challenged plants supported more bacteria than buffer-treated plants, indicating either that successful bacterial colonizers were unintentionally co-inoculated with *Hpa*, or that *Hpa* caused changes in the native flora (*Figure 5b*). We noted one outlier sample (red arrows, *Figure 5b and c*) with especially high microbial load. This sample was also an outlier by CFU counting, and because hamPCR fell within the quantitative range defined by host abundance and sequencing depth (*Appendix 1—figure 3*), we conclude that this outlier was likely not due to limitations of hamPCR.

To confirm that the abundances for all three amplicons accurately reflected the concentration of their original templates, we prepared a stepwise titration panel with real samples, mixing increasing

amounts of DNA from an uninfected *eds1-1* plant (low load) into decreasing amounts of DNA from an *Hpa*-infected *eds1-1* plant (high load). Sequencing triplicate hamPCR libraries revealed a stepwise increase in ASV levels for all amplicons, consistent with the expectation based on pipetting (*Figure 5d*). These data, combined with the infection experiment, show that hamPCR is quantitative for at least two independent microbial amplicons in real-world samples.

## Utility in diverse hosts and crops with large genomes

To demonstrate the utility of hamPCR outside of plants, we prepared samples from the nematode worm *Pristionchus pacificus*. Small hosts like nematodes where the entire individual is typically lysed during DNA preparation are ideal for hamPCR; the choice of this nematode was due to the convenience of a lab specializing in studies of this species in our institute (*Sommer, 2006*). *P. pacificus* was fed on a diet of either pure *E. coli* OP50, or alternatively a mix of *E. coli* OP50 with *Pst* DC3000 and *Lysinibacillus xylanilyticus*. The worms were washed extensively with PBS buffer to remove epidermally-attached bacteria, enriching the worms for gut-associated bacteria, and we prepared DNA from each sample. In the same manner as described in the previous section, we titrated the two DNA samples into each other to create a panel of samples representing a continuous range of colonization at biologically relevant levels. Over three replicates, hamPCR accurately captured the changing bacterial loads of the gut microbes (*Figure 5e*, *Figure 5—figure supplement 2*).

We similarly validated the technique for fungal and bacterial microbes of *Triticum aestivum* (bread wheat), because it is the most widely grown crop in the world yet one of the most difficult to study due to being hexaploid and having a 16 Gb haploid genome (*Appels et al., 2018*). To simulate different levels of infection, we titrated DNA from axenically-grown wheat leaves into DNA from wheat roots that had been cultivated in non-sterile soil and applied hamPCR, using as a host gene RNA polymerase A1 (*PolA1*), which is present as a single copy in each of the A, B, and D subgenomes (*Rai et al., 2012*). We recovered expected abundance patterns in the panel both for ITS1 rDNA primers (*Figure 5f*, *Figure 5—figure supplements 3* and *4*) and for V4 16S rDNA primers (*Figure 5—figure supplements 3* and *4*). We noticed the original ratio of ITS1 to *PolA1* sequences recovered was low; because ITS primers produce amplicons that are highly variable in length, some of which may co-migrate with the host amplicon on a gel, the cut-and-mix approach described in *Figure 3* could not be used to improve ITS1 representation. However, increasing the ratio of the ITS1:*PolA1* HM-tagging primers from 1:1 to 2:1 (Materials and methods) successfully enriched the ITS1 amplicon without sacrificing relative load determination between samples (*Figure 5—figure supplements 3* and *4*).

To go beyond model organisms and controlled titrations and to demonstrate the ability of hamPCR to yield biological insights into complex study systems, we conducted two additional experiments with crop plants. First, we set up a growth curve in bell pepper (*Capsicum annuum*), which has a 3.5 Gb genome (*Kim et al., 2014*), approximately 25× larger than *A. thaliana*, and the pepper pathogenic bacterium *Xanthomonas euvesicatoria* (*Xe*) strain 85-10 (*Thieme et al., 2005*). As proof-of-concept preparation for the growth curve, to confirm that hamPCR could accurately capture absolute changes in pathogen abundance in pepper leaves, we constructed an infiltration panel in which *Xe* 85–10 was diluted to final concentrations of $10^4$, $10^5$, $10^6$, $10^7$, and $10^8$ CFU / mL and infiltrated into four replicate leaves per concentration. Immediately afterwards, without further bacterial growth, we harvested leaf discs within inoculated areas using a cork borer. We ground the discs and used some of the lysate for *Xe* 85–10 CFU counting, and the remainder for DNA extraction and hamPCR targeting the V4 16S rDNA and the pepper *GI* gene (*CaGI*), and qPCR, targeting the *xopQ* gene for a *Xe* type III effector (*Doddaraju et al., 2019*), and the *C. annuum UBI-3* gene for a ubiquitin-conjugating protein (*Wan et al., 2011*).

Sequencing the hamPCR libraries revealed that as the *Xe* 85–10 infiltration concentration increased, so did the resulting load of the ASV corresponding to *Xe* 85–10 (*Figure 6—figure supplement 1* and *Figure 6—figure supplement 2a*). The other major bacterial classes detected in the infiltration panel, comprising commensal bacteria already present in the leaves, had similar, low abundances, regardless of the amount of infiltrated *Xe* 85–10 (*Figure 6—figure supplement 2b*). When we scaled hamPCR and qPCR values to fit the range of CFU counts recovered from the same lysates (Materials and methods), the hamPCR and qPCR *Xe* 85–10 ASV loads showed nearly the same exponential differences between samples, although at lower infiltration concentrations, qPCR and hamPCR gave a slightly higher estimate than CFU counts (*Figure 6—figure supplement 2c*).

The presence of a low level of native, antibiotic-sensitive *Xe* on the leaves could potentially explain this discrepancy, because this could be detected by DNA-based methods but not culturing.

For the pepper growth-curve, we infiltrated six *C. annuum* leaves of six different plants with *Xe* 85–10 at a concentration of $10^4$ CFU / mL, and took samples from each plant at 0, 2, 4, 7, 9, and 11 dpi for CFU counting, qPCR, and hamPCR. We observed a rapid increase in *Xe* 85–10 ASV abundance as a result of rapid bacterial growth, leveling off at 7 dpi (*Figure 6a*). By 7 dpi, bacterial growth had reduced the host *GI* amplicon abundances for most samples to levels at which load calculations become less reliable at their sequencing depth (as defined in *Appendix 1—figure 3d*); this was also the case at 9 and 11 dpi (gray box, *Figure 6a*). Scaled *Xe* 85–10 ASV loads compared very closely to CFU counts and to scaled qPCR abundances up to 7 dpi (*Figure 6c*). Notably, the other major bacterial classes, Actinobacteria and the Alpha-, Beta-, and Gammaproteobacteria, also increased in microbial load through time, a trend significant even comparing 2 dpi to 0 dpi (*Figure 6b*, Mann-Whitney U-test, $p < 0.001$). This increase in load for the other classes was not a PCR artifact due to high *Xe* 85–10 titers, because in the infiltration panel, measurements for these classes had not changed even at higher pathogen concentrations (*Figure 6—figure supplement 2b*). This subtle but biologically significant effect of infection on growth of commensal bacteria would be completely invisible in a pure 16S rDNA amplicon analysis, which would only show *Xe* 85–10 overtaking the community.

Finally, we applied hamPCR to DNA from 201 leaf samples from mature, isogenic maize (B73) growing in a field site in Tübingen, Germany. We used the V4 region of 16S rDNA for bacteria and the single copy *LUMINIDEPENDENS* (*LD*) gene as a host marker, and plotted both the relative abundance of bacterial genera (*Figure 6d*) and the bacterial load of these genera (*Figure 6e*). In some samples, the genus *Sphingomonas* exceeded 80% of the bacterial community, creating especially strong compositionality effects; other abundant genera *Perlucidibaca, Limnohabitans,* and *Acidovorax* visibly increased in relative abundance as *Sphingomonas* became less abundant (*Figure 6d*). In contrast, the bacteria load of these same genera appeared mostly unaffected by *Sphingomonas* bacterial load (*Figure 6e*). As expected, a Pearson correlation network made with relative abundance data revealed that *Sphingomonas* was negatively correlated with many genera, a well-known and problematic artifact of compositionality (*Friedman and Alm, 2012*; *Figure 6f*). A Pearson correlation network made with microbial load data was remarkable in that *Sphingomonas,* despite having the highest median abundance of any genus, is among the genera least correlated with others (*Figure 6f*). We also calculated a correlation network using SparCC (*Friedman and Alm, 2012*), which estimates Pearson correlations on log-transformed components to avoid compositionality artifacts. This network did indeed avoid the spurious negative correlations with *Sphingomonas,* although it still implicated the genus more strongly than the correlation network built with hamPCR data. Each network has a very different biological interpretation, with negative Sphingomonas correlations implying that the genus as a whole can greatly influence the colonization of other microbes. The weak connectedness of Sphingomonas in the microbial load data does not necessarily imply that Sphingomonas strains do not influence colonization of other microbes, but rather that such effects, if they exist, can not be inferred from the abundance of the genus as a whole. Future study will be necessary to resolve these issues. Overrepresentation of a few abundant organisms is a feature shared by most ecological communities (*McGill et al., 2007*), including microbial communities (*Zhou et al., 2013*); overcoming this compositionality problem is broadly relevant to studies of host-associated microbiomes.

## Discussion

We developed hamPCR, a simple and robust method to quantitatively co-amplify one or more microbial marker genes along with an unrelated host gene, allowing accurate determination of microbial load and microbial community composition from a single sequencing library (*Figure 1*, *Figure 2*). Furthermore, we developed a method to predictably optimize the amount of sequencing effort devoted to microbe vs. host, without losing information about the original microbe to host ratio (*Figure 3*). This is an important advance in our approach that greatly increases cost-efficiency.

The principle behind hamPCR stands on a body of literature describing related, firmly established techniques, which bodes well for wide-spread adoption of our approach. Using two steps in a PCR protocol is common in amplicon sequencing, including of microbial marker genes (*de Muinck et al.,*

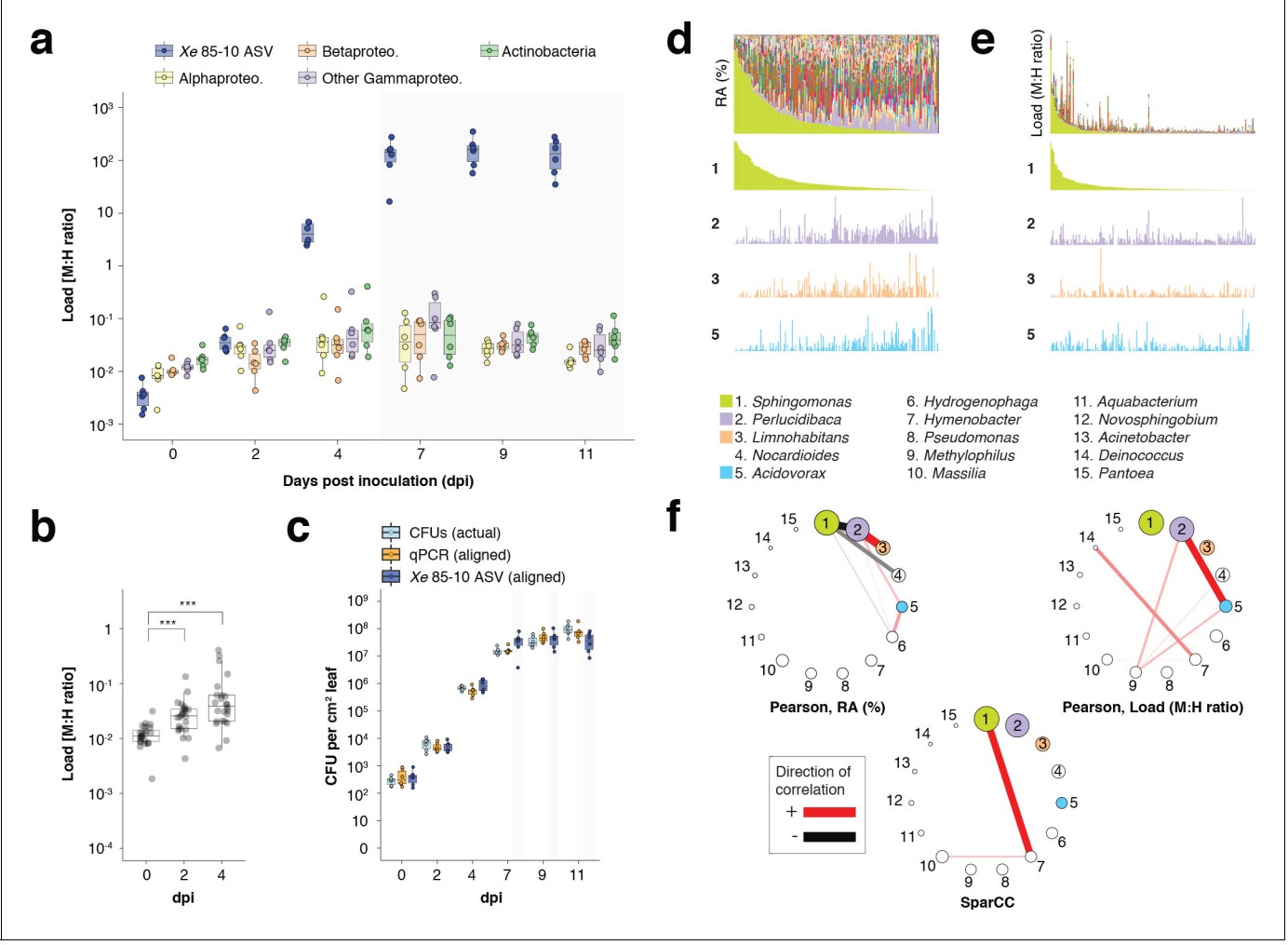

**Figure 6.** hamPCR can provide new insights into microbial interactions in crop plants. (a–c) *C. annuum* growth curve experiment. All y-axes are on a base-10 logarithmic scale. In all boxplots, the median is represented by a horizontal line and box boundaries show the lower and upper quartiles. Whiskers extend from the box up to 1.5 times the interquartile range. (a) *Xe* 85–10 was inoculated into *C. annuum* leaves at $10^4$ CFU/mL. Leaf samples were taken at 0, 2, 4, 7, 9, and 11 days post inoculation (dpi), and hamPCR performed. The corrected load is shown for the particular ASV corresponding to *Xe* 85–10, as well as for the major bacterial classes. (b) The total load for all bacterial classes shown in a at 0, 2, and 4 dpi (***$p<0.001$, Mann-Whitney U-test). (c) Actual CFU counts for *Xe* 85–10 in the growth curve experiment juxtaposed with scaled qPCR and hamPCR loads. (d-e) Field-grown *Z. mays* collection. (d) Relative abundance (RA) of bacterial genera found in *Z. mays* leaf hole punches, ordered by *Sphingomonas* relative abundance. The genera corresponding to the first 15 colors from bottom to top are shown in reverse order in the legend. The relative abundance of four isolated genera is highlighted (colored boxes in legend). (e) Same as (d) but showing microbial load rather than relative abundance and ordered by *Sphingomonas* load. (f) Correlation networks of the same 15 genera from the legend for d and e. Pearson correlation from RA data from d (left), pearson correlation of microbial load from e (right), and SparCC correlation network (bottom). Circles representing genera are scaled such that their area represents the median genus abundance across all samples. Only correlations of absolute magnitude >= 0.3 are shown.

The online version of this article includes the following figure supplement(s) for figure 6:

**Figure supplement 1.** Gels from *C. annuum* experiments.

**Figure supplement 2.** hamPCR can be generalized to Xanthomonas infection of *C. annuum*.

*2017*; *Gohl et al., 2016*; *Holm et al., 2019*; *Lundberg et al., 2013*; *Symeonidi et al., 2020*). Two-step PCR protocols provide the major advantage that only a one-time investment is needed in a set of universal barcoding primers for a flexible step two. These can be easily adapted to any amplicon (s) by simply swapping in different template-specific primers for step one. For labs already equipped for two-step PCR, implementing hamPCR involves only slight adjustments to cycling conditions and template-specific HM-tagging primers (Appendix 1 - Discussion 1).

Quantitative co-amplification using multiple primer pairs also has proven reliable (*Carlson et al., 2013*; *Weller et al., 2000*; *Wen and Zhang, 2012*), and PCR biases affecting co-amplification of diverse fragments are manageable and well-understood from popular RNA-seq and WMS protocols (*Bowers et al., 2015*; *Rinke et al., 2016*). This rich literature should increase confidence when implementing hamPCR in microbiome research, and it also provides resources for optimization and further development. For example, the use of fewer cycles in exponential PCR could reduce noise and bias, hamPCR HM-tagging primers could be fitted with unique molecular identifiers (UMIs) for higher precision, and the protocol could be adapted for sequencing platforms with longer read lengths.

We have demonstrated that microbial load measurement is sensitive to the relative concentrations between the host and microbe primers in the HM-tagging step (*Appendix 1—figure 4*), consistent with the effects of primer concentration on amplification efficiency in qPCR (*Bustin and Huggett, 2017*; *Pierce et al., 2005*; *Sanchez et al., 2004*). This property makes it possible to fine-tune the primer ratios, either to yield the expected ratio of products (*Carlson et al., 2013*), or to intentionally increase the representation of a microbial amplicon for more efficient sequencing (*Figure 5—figure supplement 4a–d*). The effect of primer concentration has important implications for how a large project should be prepared. We recommend that the HM-tagging primers be carefully pipetted into a multiplexed primer master mix sufficiently large to be used for the entire project, or alternatively the same control samples should be sequenced across sample batches to allow correction of slight batch differences.

Because hamPCR can only quantify the DNA available in the template, choice of sample and appropriate DNA extraction methods are very important. In particular, the sample must in the first place include a meaningful quantity of host DNA. For example, although there is some host DNA in mammalian fecal samples or in plant rhizosphere soil samples, this host DNA does not accurately represent the sample volume, and therefore relating microbial abundance to host abundance probably has less value in these cases. Further, the DNA extraction method chosen must lyse both the host and microbial cell types. An enzymatic lysis suitable for DNA extraction from pure cultures of *E. coli* may not lyse host cells or even other microbes. Appropriate DNA preparation methods for metagenomics have been thoroughly evaluated elsewhere (*Albertsen et al., 2015*; *Yuan et al., 2012*), and a common point of agreement is that strong bead-beating increases the yield and completeness of the DNA extraction, but comes at the cost of some DNA fragmentation. Especially for short reads, as we have used here, this fragmentation is not a problem, and we recommend to err on the side of a harsher lysis, using strong bead beating potentially preceded by grinding steps using a mortar and pestle as necessary for tougher tissue.

A limitation of hamPCR is reduced accuracy at the highest microbial loads (*Appendix 1—figure 3*). Only a minority of our samples reached a level of infection that interfered with accurate quantification, and we expect that this will be the case for most colonized hosts. If not, there are three straightforward adjustments that can increase host signal to acceptable levels. First, altering the host and microbe amplicon ratio in the pooled library prior to sequencing, as demonstrated in *Figure 3*, could be used to *increase* the overall host representation. Second, a host gene with a higher copy number could be chosen for HM-tagging throughout the entire project, which would increase host representation by a factor of that copy number (*Kembel et al., 2012*). Finally, adjusting the concentration of the host primers in the HM-tagging reaction could also increase the representation of host (*Appendix 1—figure 4*; *Carlson et al., 2013*). hamPCR is best suited to comparing microbial loads within an experimental system using consistent host and microbe primers, because different microbe primer pairs can differ in their amplification efficiency and therefore yield different load measurements on the same template DNA (*Figure 4*). Additionally, researchers may decide to adjust primer ratios for more efficient sequencing, which can also change the calculated load. However, looking toward future best practices, this lack of cross-comparability can be overcome by including a simple standard curve. This can be prepared similarly to our synthetic titration panel in *Figure 1* which used pure plant DNA mixed with bacterial DNA. Alternatively, a pure host amplicon can be mixed with a pure microbial amplicon at different known ratios. By using hamPCR to sequence standards along with experimental samples, the measured microbial loads can be correlated with known 'standardized' load ratios, which in turn can be compared across primer sets and labs.

In summary, we have demonstrated that hamPCR is agnostic to the taxonomic identities of the organisms studied on both the host and microbe side, their genome sizes, or the functions of the regions amplified. We have also shown that hamPCR can monitor three amplicons at the same time

for interkingdom microbial quantification, and in principle can multiplex more with careful design. Our focus here has been on tracking hosts and their closely-associated microbes, but the protocol could also be adapted to quantitatively relate different amplicons targeting archaea, bacteria, and fungi in diverse 'host-free' environments like soil. Besides whole organisms, hamPCR also enables quantitative monitoring of bacterial populations and sub-genomic elements, such as plasmids or pathogenicity islands that might not be shared by all strains in a population. An exciting application of hamPCR is the study of endophytic microbial colonization and infection in crop plants, many of which have very large genomes that preclude the analysis of any sizable number of samples by WMS. In a previous study, we sequenced leaf metagenomes from over 200 *A. thaliana* plants, at not insignificant costs (*Regalado et al., 2020*). In wheat, assuming comparable microbial loads, the same investment in sequencing would barely be sufficient for two samples due to the size of the wheat genome of over 16 Gb. WMS of the >200 samples we processed of field-grown maize likewise would be prohibitively expensive, and supplementing these data with an orthogonal method on this scale requires at least double the number of samples and double the experimental time.

Other exciting applications are the recognition of cryptic infections (*Stergiopoulos and Gordon, 2014*), tracking of mixed infections, and measurement of pathogen abundances on hosts showing quantitative disease resistance – this could even be accomplished by spiking hamPCR amplicons into the same sequencing run used to genotype the hosts (*St Clair, 2010*). In sparsely colonized samples, hamPCR will help prevent inflating the abundance of ultra-low abundance microbes, such as reagent contaminants. Finally, for projects with many samples, the fact that hamPCR derives microbial composition and load from the same library not only saves costs and uses less of the sample, but also simplifies analysis and project organization.

## Materials and methods

### Key resources table

| Reagent type (species) or resource | Designation | Source or reference | Identifiers | Additional information |
|---|---|---|---|---|
| Sequence-based reagent | HM-tagging primers | Eurofins; this paper | | Standard desalting; for sequences see *Supplementary file 1*, Appendix 1 |
| Sequence-based reagent | PCR forward primer | Eurofins; https://doi.org/10.1017/qpb.2020.6 | PCR_F_G-40610 | 5'-AATGA TACGGCGACCACCGA GATCTACACTCTTTCCC TACA CGACGCTCTTC-3'; HPLC purified |
| Sequence-based reagent | PCR reverse primers | IDT; https://doi.org/10.1038/nmeth.2634 | | IDT Ultramers; for sequences see *Supplementary file 1* |
| Sequence-based reagent | XopQ *Xanthomonas* qPCR primers | Eurofins; https://doi.org/10.1038/s41598-019-46588-9; | XopQ_F, XopQR | 5'-GCGAGGAACTTGGAA TGCTC-3' 5'-AGGCCGAAGGC TTTTTGCG-3'; Standard desalting |
| Sequence-based reagent | UBI3 *C. annuum* qPCR primers | Eurofins; https://doi.org/10.1016/j.bbrc.2011.10.105 | UBI3_F, UBI3_R | 5'-TGTCCATCTGCTCTC TGTTG-3' 5'-CACCCCAAGCACAA TAAGAC-3'; Standard desalting |
| Peptide, recombinant protein | Taq DNA Polymerase | NEB | Cat. #: M0267 | |
| Peptide, recombinant protein | Q5 DNA Polymerase | NEB | Cat. #: M0491 | |

*Continued on next page*

*Continued*

| Reagent type (species) or resource | Designation | Source or reference | Identifiers | Additional information |
|---|---|---|---|---|
| Peptide, recombinant protein | pPNA, mPNA | PNAbio | Cat. #: PP01, MP01 | Peptide nucleic acids for blocking organelle amplification when using hamPCR on plants. |
| Software, algorithm | USEARCH | www.drive5.com | version 11 | |
| Recombinant DNA reagent | Synthetic equimolar plasmid template (plasmid) | This paper | | For sequence and construction, see Appendix 1 - Discussion 2. |
| Other | Solid phase reversible immobilization (SPRI) beads | https://dx.doi.org/10.1101%2Fgr.128124.111 | | |

## hamPCR protocol

hamPCR requires two steps: a short 'HM-tagging' reaction of two cycles, and a longer 'exponential' reaction. We used 30 cycles throughout this work, although fewer can and should be used if the signal is clear for better quantitative results. The primers employed in the HM-tagging reaction were used at ⅛ the concentration of the exponential primers, as this still represents an excess in a reaction run for only two cycles, prevents waste, and reduces dimer formation. See Appendix 1 and *Supplementary file 1* for detailed information about the primers. In particular, Appendix 1 - Discussion 1 provides guidance on adapting HM-tagging primers to other two-step PCR protocols, and Appendix 1 - Discussion 3 discusses the design of new HM-tagging primers.

### HM-tagging reaction

We used Taq DNA Polymerase (NEB, Ipswich, MA, USA) for the first HM-tagging step, and set up 25 µL reactions as follows.

| Master Mix | For 25 µL |
|---|---|
| 10x Taq buffer | 2.5 µL |
| Taq polymerase | 0.2 µL |
| dNTPs (10 mM) | 0.5 µL |
| * HM-tagging primer mix | 1.25 µL |
| ** PNAs (mix of mPNA and pPNA each at 50 µM) | 0.375 µL |
| *** DNA (2–10 ng/µL) | 5 µL |
| Water | (to 25 µL) |

\* HM-tagging primer mix is an equimolar mix of HM-tagging primers with each at a partial concentration of 1.25 µM.

\*\* PNA was used to block chloroplast and mitochondrial amplification in reactions involving the V4 or V3V4 region of 16S rDNA. For other reactions, it is not helpful and is omitted.

\*\*\* not part of master mix.

Each well received 20 µL of master mix and 5 µL of DNA (around 50 ng). Completed reactions were thoroughly mixed on a plate vortex and placed into a preheated thermocycler. We used the following standard cycling conditions:

1. 94°C for 2 min. *Denature*
2. 78°C for 10 s. *PNA annealing*
3. 58°C for 15 s. *Primer annealing*
4. 55°C for 15 s. *Primer annealing*
5. 72°C for 1 min. *Extension*
6. GO TO STEP 1 for 1 additional cycle
7. 16°C forever *Hold*

The HM-tagging reaction was cleaned with Solid Phase Reversible Immobilization (SPRI) beads (*Rohland and Reich, 2012*). All ITS amplicons were cleaned with a 1.1:1 ratio of SPRI beads to DNA, or 27.5 µL beads mixed in 25 µL of tagged template. After securing beads and DNA to a magnet and removing the supernatant containing primers and small fragments, beads were washed twice with 80% ethanol, air dried briefly, and eluted in 17 µL of water. For primer sequences, see Appendix 1 and *Supplementary file 1*.

## Exponential reaction

Of the tagged DNA from step one, 15 µL was used as template for the exponential reaction. To reduce errors during the exponential phase, we used the proof-reading enzyme Q5 from NEB, with its included buffer. We prepared reactions in 25 µL for technical tests with replicated samples. For samples prepared without sequenced replicates, we prepared most in triplicate reactions in which a 40 µL mix was split into three parallel reactions of ~13 µL prior to PCR to reduce bias, although this is likely unnecessary (*Marotz et al., 2019*).

| Master Mix | For 25 µL | For 40 µL |
|---|---|---|
| 5x Q5 buffer | 5 µL | 8 µL |
| Q5 polymerase | 0.25 µL | 0.4 µL |
| dNTPs (10 mM) | 0.5 µL | 0.8 µL |
| 100 µM F universal PCR primer | 0.0625 µL | 0.1 µL |
| * PNAs (mix of mPNA and pPNA each at 50 µM) | 0.375 µL | 0.6 µL |
| ** 5 µM reverse barcoded primer | 1.25 µL | 2 µL |
| ** DNA (from previous reaction) | 15 µL | 15 µL |
| Water | (to 25 µL) | (to 40 µL) |

*PNA was used to block chloroplast and mitochondrial amplification in reactions involving the V4 or V3V4 region of 16S rDNA. For other reactions, it is not helpful and is omitted.

** not part of master mix.

We first distributed 8.75 µL (or 23 µL for 40 µL mixes) of master mix to each well. We then added 15 µL of the DNA from the HM-tagging reaction and 1.25 µL (or 2 µL for 40 µL mixes) of 5 µM barcoded reverse primer. For the 40 µL mixes, 13 µL was pipetted into two new PCR wells. The PCR reactions were placed into a hot thermocycler and cycled with the following standard conditions:

1. 94°C for 2 min. *Denature*
2. 94°C for 20 s. *Denature*
3. 78°C for 5 s. *PNA annealing*
4. 60°C for 30 s. *Primer annealing*
5. 72°C for 45 s. *Extension*
6. GO TO STEP 2 for 29 additional cycles
7. 16°C forever *Hold*

Following PCR, sets of three 13 µL reactions were recombined to 40 µL. For primer sequences, see Appendix 1 and *Supplementary file 1*.

## Library quality control and pooling

For visualization, 5 µL of PCR product was mixed with 3 µL of 6x loading dye and all 8 µL loaded on a 2% agarose gel and stained with ethidium bromide. The remaining PCR products were cleaned with a SPRI-to-DNA ratio of 1.1:1.0 (v/v). The DNA concentrations in the cleaned products were measured with PicoGreen (Invitrogen, Carlsbad, CA, USA) and samples were pooled at equimolar total DNA ratios. We note that because host and microbial fractions are independently visible on the gel, it would also be possible to measure the quantity of microbial products with image analysis software such as ImageJ (*Rueden et al., 2017*) and pool at equimolar microbial ratios.

The pooled library was diluted to ~1 ng/µL and run on a Bioanalyzer High Sensitivity DNA chip (Agilent, Santa Clara, CA, USA) to check library purity and to estimate the expected ratio of host to microbial amplicons in the sample.

## Pre-sequencing adjustment of host: microbe ratio

To adjust the host-to-microbe ratio in the 'synthetic template panel' and 'cycle number test' prior to sequencing on a HiSeq3000 instrument (Illumina, San Diego, CA, USA), four reference samples were first made by rebarcoding the original pooled library (*Figure 3—figure supplements 1* and *2*). To accomplish this, ~5 ng of of the pooled library was used in a 30 µL PCR reaction as follows:

| Master Mix | For 30 µL |
|---|---|
| 5x Q5 buffer | 6 µL |
| Q5 polymerase | 0.3 µL |
| dNTPs (10 mM) | 0.6 µL |
| 100 µM F universal PCR primer | 0.075 µL |
| ** 5 µM reverse barcoded primer | 1.5 µL |
| ** 5 ng original pooled library | 5 µL |
| Water | 16.45 µL |

** not part of master mix.

After distributing 23.5 µL of master mix to each well, 5 µL of the diluted original library was added to each well (5 ng total), along with 1.5 µL of 5 µM barcoded reverse primer. Just prior to placing the reactions in the thermocycler, a 5 µL pre-PCR aliquot was removed from each one and kept on ice to preserve the pre-PCR concentrations. The remaining 25 µL reaction was placed into a preheated thermocycler and run for eight cycles, using the following cycling conditions:

1. 94° C for 2 min. *Denature*
2. 94° C for 30 s. *Denature*
3. 78° C for 5 s. *PNA annealing*
4. 60° C for 1 min. *Primer annealing*
5. 72° C for 1 min. *Extension*
6. GO TO STEP 2 for seven additional cycles
7. 16° C forever *Hold*

Following PCR, the pre-PCR aliquots were run alongside 5 µL of post-PCR product on a 2% gel to confirm successful amplification of the reference libraries (*Figure 3—figure supplement 2*). The remaining 20 µL of PCR reactions were then cleaned with SPRI beads (1.5: 1.0 [v/v]) and set aside. A large aliquot of the original library (approximately 50 ng) was also run on a 2% gel to separate the host and microbe bands for individual purification. The bands were cut out of the gel and each band was put into a separate Econospin spin column (Epoch Life Sciences, Missouri City, TX, USA) without any other liquids or binding buffer. The gel slices were centrifuged at maximum speed to force the liquid containing the DNA into the bottom chamber, leaving the dried gel on top. The eluted DNA was cleaned with SPRI beads at 1.5: 1.0 (v/v) and eluted in EB.

The purified pooled host library fraction, pooled microbe library fraction, and each of the four reference libraries were quantified with Picogreen and the molarity of each was estimated. The pools were then mixed together, targeting host molarity at 5% of the total and each reference library at 1% of the total.

## Illumina sequencing

Pooled and quality-checked sequencing libraries were cleaned of all remaining dimers and off-target fragments using a BluePippin (Sage Science, Beverly, MA, USA) set to a broad range of 280 to 720 bp. The libraries were then diluted for Illumina sequencing following manufacturers' protocols. Libraries were first diluted to 2.5–2.8 nM in elution buffer (EB, 10 mM Tris pH 8.0) and spiked into a compatible lane of the HiSeq3000 instrument (2 x 150 bp paired end reads) to occupy 2–3% of the lane. Samples were sequenced across four total lanes (*Supplementary file 1*).

## Sequence processing

The sequences were demultiplexed first by the 9 bp barcode on the PCR primers (*Supplementary file 1*), of which there are 96, not allowing for any mismatches. In some cases in

which two samples differed in both their host and microbe primer sets, we amplified both samples with the same 9 bp barcode to increase multiplexing; such samples were further demultiplexed using regular expressions for the forward primer and reverse primer sequences. Following demultiplexing, all samples were filtered to remove sequences with any mismatches to the expected primers. With HiSeq3000 150 bp read lengths, overlap of read 1 and read 2 was not possible for our amplicons, and therefore only read 1 was processed further.

All primer sequences were removed. Additional quality filtering, removal of chimeric sequences, preparation and Amplicon Sequence Variant (ASV) tables, and taxonomic assignment were done with a combination of VSEARCH (*Rognes et al., 2016*) and USEARCH10 (*Edgar, 2010*). ASVs were prepared as 'zero-radius OTUs' (zOTUs) (*Edgar, 2010*). The 16S rDNA taxonomy was classified based on the RDP training set v16 (13 k seqs.) (*Cole et al., 2014*), and ITS1 taxonomy of the top 10 most abundant fungal ASVs was classified manually using the UNITE database (*Nilsson et al., 2019*) (https://unite.ut.ee/). To reduce memory usage, data from the five lanes was processed into four independent ASV tables (Supplementary Data), as described in the sample metadata (*Supplementary file 1*).

ASV tables were analyzed statistically and graphically using custom scripts in R (*R Development Core Team, 2019*), particularly with the help of packages 'ggplot2' (*Wickham, 2016*) and 'reshape2' (*Wickham, 2007*). Custom scripts are available on GitHub at (https://github.com/derekLS1/hamPCR).

## Samples
### Synthetic titration panel
Seeds from the *Arabidopsis thaliana* accession Col-0 were surface sterilized by immersion for 1 min in 70% ethanol with 0.1% Triton X-100, soaking in 10% household bleach for 12 min, and washing three times with sterile water. Seeds were germinated axenically on ½ strength MS media with MES, and about 2 g of seedlings were harvested after 10 days. DNA was extracted in the sterile hood as in *Regalado et al., 2020* and diluted to 10 ng/µL in elution buffer (10 mM Tris pH 8.0, hereafter EB). Pure *E. coli* and *Sphingomonas* sp. cultures were likewise grown with LB liquid and solid media respectively, and DNA was extracted using a bead beating protocol (*Regalado et al., 2020*). *E. coli* DNA was used separately, or alternatively pooled with the mixed *Sphingomonas* DNA, and diluted to 10 ng/µL. The plant DNA and microbial DNA were then combined according to the following table:

| | 100E | 100 | 24 | 20 | 16 | 8 | 4 | 2 | 1 | 0.5 | 0.25 | 100P | 100P | blank |
|---|---|---|---|---|---|---|---|---|---|---|---|---|---|---|
| µL *A. thaliana* DNA (10 ng/µL) | | | 760 | 800 | 840 | 920 | 960 | 980 | 990 | 995 | 997.5 | 1000 | 1000 | |
| µL mixed bacterial DNA (10 ng/µL) | | 1000 | 240 | 200 | 160 | 80 | 40 | 20 | 10 | 5 | 2.5 | | | |
| µL *E. coli* DNA (10 ng/µL) | 1000 | | | | | | | | | | | | | |
| µL water | | | | | | | | | | | | | | 1000 |

V4 HM-tagging was performed with 515_F1_G-46603 and 799_R1_G-46601 (V4 16S rDNA) and At.GI_F1_G-46602 and At.GI_R502bp_G-46614 (*A. thaliana GI*). Each exponential PCR reaction was completed in a single reaction of 25 µL.

### Synthetic equimolar plasmid template
The ITS1 region from *Agaricus bisporus*, a fragment of the *GI* gene from *A. thaliana* Col-0 accession, the 16S rRNA gene from *Pst* DC3000, and the ITS1 region from *H. arabidopsidis* were PCR amplified individually, combined into one fragment via overlap extension PCR, and cloned into pGEM-T Easy (Promega, Madison, WI, USA). The sequences of these templates can be found in Appendix 1 – Discussion 2.

### Wild *A. thaliana* samples
DNA from chosen samples previously analyzed by conventional 16S rDNA-only sequencing of the V4 region and WMS (*Regalado et al., 2020*) was reused, chosen to capture a wide range of realistic bacterial loads. The samples were individually assayed with hamPCR using 5 µL DNA template

(approximately 50 ng). V4 HM-tagging was performed with 515_F1_G-46603 and 806_R1_G-46631 (V4 16S rDNA) and At.GI_F1_G-46602 and At.GI_R502bp_G-46614 (502 bp *A. thaliana GI*). These were the only V4 samples tagged with the 806R primer instead of the nearby 799R primer, and it was used to enable direct comparison to the dataset in *Regalado et al., 2020*. V3V4 HM-tagging was performed with 341_F1_G-46605 and 799_R1_G-46601 (V3V4 16S rDNA) and At.GI_F1_G-46602 and At.GI_R502bp_G-46614 (502 bp *A. thaliana GI*). V5V6V7 HM-tagging was performed with 799_F1_G-46628 and 1192_R1_G-46629 (V5V6V7 16S rDNA) and At.GI_F1_G-46602 and At. GI_R466bp_G-46652 (466 bp *A. thaliana GI*). Each exponential PCR reaction was completed in a single reaction of 25 μL; each sample was replicated three times.

## Wild *A. thaliana* mixed sample

DNA from samples previously analyzed by conventional 16S rDNA-only sequencing of the V4 region and WMS (*Regalado et al., 2020*) were pooled to prepare a single abundant mixed sample to be used repeatedly for technical tests.

## *Hyaloperonospora arabidopsidis* and *Pseudomonas syringae pv. tomato* DC3000 co-infection

Both wildtype *A. thaliana* seedlings in the Col-0 genetic background and *enhanced disease susceptibility one* mutants in the Ws-0 genetic background (*eds1-1*) were grown from surface-sterilized seeds. Seedlings were raised in ED73 potting mix (Einheitserdewerke, Sinntal-Altengronau, Germany) in 5 cm pots for 10 days under short-day conditions (8 hr light, 16 hr dark). Each pot contained four to ive seedlings, and for each genotype, four pots were used for each infection condition. Plants were treated with either 10 mM MgCl$_2$ (buffer only), *H. arabidopsidis* (*Hpa*) isolate 466–1 alone (5 x 10$^4$ spores / mL), or *Hpa* 466–1 with *P. syringae pv. tomato* (*Pst*) DC3000 (OD600 = 0.25, a gift from El Kasmi lab, University of Tübingen).

The infected plants were grown at 16°C for 8 days (10 hr light, 14 hr dark) and harvested by pooling all seedlings in each pot into a sterile pre-weighed tube, which was again weighed to find the mass of the seedlings. Three 5 mm glass balls and 300 μL 10 mM MgCl$_2$ were added to each tube and the plant cells were lysed at a speed of 4.0 m/s for 20 s in a FastPrep-24 Instrument (MP Biomedicals, Illkirch-Graffenstaden, France) to release the live bacteria from the leaves. From the pure lysate, 20 μL was used for a serial log dilution series, and 5 μL of each dilution was plated on LB agar supplemented with 100 μg/mL rifampicin. Colony-forming units (CFUs) were counted after 2 days of incubation at 28°C. The remaining 280 μL of lysate were combined with 520 μL DNA lysis buffer, 0.5 mL of 1 mm garnet sharp particles (BioSpec, Bartlesville, OK, USA). Of 20% SDS, 60 μL was added to make a final SDS concentration of 1.5%, and DNA was extracted using a bead beating protocol (*Regalado et al., 2020*). The number of *Hpa* sporangiophores was too high to be accurately quantified by visual counting.

The DNA preps were individually assayed with hamPCR using 5 μL DNA template (approximately 30 ng); HM-tagging was performed with three primer sets: Ha.Actin_F1_G-46716 and Ha. Actin_R1_G-46717 (*Hpa* Actin), At.GI_F1_G-46602 and At.GI_R502bp_G-46614 (502 bp *A. thaliana GI*), and 515_F1_G-46603 and 799_R1_G-46601 (V4 16S rDNA). Each exponential PCR reaction was completed in three parallel reactions of 13 μL, which were recombined prior to sequencing.

## Titration with plant DNA infected with *H. arabidopsidis*

A titration panel was made combining different amounts of DNA from uninfected plants (*eds1-1* treated only with 10 mM MgCl$_2$) and DNA from *Hpa*-infected plants (*eds1-1* infected with *Hpa* as described above). Infected and uninfected pools were each diluted to 6 ng/μL, and combined in 0:7, 1:6, 2:5, 3:4, 4:3, 5:2, 6:1, and 7:0 ratios. These were tagged using the same three primer sets described above for *Hpa* actin, *A. thaliana GI*, and V4 16S rDNA above. Each exponential PCR reaction was completed in a single reaction of 25 μL; hamPCR was replicated on the titration three times.

## *Capsicum annuum* infections with *Xanthomonas*
### Leaf infiltration log series

Using pressure infiltration with a blunt-end syringe, *C. annuum* cultivar Early Calwonder (ECW) leaves were inoculated with *Xanthomonas euvesicatoria* (*Xe*). *Xe* strain 85–10 (*Thieme et al., 2005*) was resuspended in 10 mM $MgCl_2$ to final concentration of $10^8$ CFU / mL ($OD_{600}$=0.4) and further diluted to $10^7$, $10^6$, $10^5$, and $10^4$ CFU / mL. Upon infiltration, five leaf discs (7 mm diameter) were punched from each leaf per sample and placed in a 2 mL round-bottom tube with two SiLibeads (type ZY-S 2.7–3.3 mm, Sigmund Lindner GmbH, Warmensteinach, Germany) and 300 µL 10 mM $MgCl_2$. The samples were ground by bead beating for 25 s at 25 Hz using a Tissue Lyser II machine (Qiagen, Hilden, Germany). For CFU-based bacterial enumeration, 30 µL of the lysate or 30 µL of serial dilutions were plated on NYG medium (0.5% peptone, 0.3% yeast extract, 0.2% glycerol and 1.5% agar [*Daniels et al., 1984*] containing rifampicin 100 µg/ml). *Xe* bacteria were counted 3 days post incubation at 28˚C. The remaining 250 µL of lysate was combined with 600 µL of DNA lysis buffer containing 2.1% SDS (for a 1.5% final SDS concentration) and transferred to screw cap tubes filled with 1 mm garnet sharp particles, for a bead-beating DNA prep as previously described (*Regalado et al., 2020*).

*Growth curve: Xe* strain 85–10, resuspended in 10 mM $MgCl_2$ to a final concentration of $10^4$ CFU / mL, was infiltrated via a blunt end syringe into 6 *C. annuum* (ECW) leaves of six different plants. Upon 0, 2, 4, 7, 9, and 11 dpi (days post inoculation) four leaf discs (7 mm diameter) from each inoculated leaf were harvested and bacterial numbers were determined as described above. Of leaf lysates, 250 µL were used for a bead-beating DNA prep as described for all other samples above.

Each hamPCR template HM-tagging reaction used 5–10 µL template (approximately 50 ng each); HM-tagging was performed with primers 515_F3_G-46694 and 799_R1_G-46601 (V4 16S rDNA), and Ca.GI_F1_G-46626 and Ca.GI_R1_G-46627 (*C. annuum GI*). Each exponential PCR reaction was completed in three parallel reactions of 13 µL, which were recombined prior to sequencing.

## *Pristionchus pacificus* titration panel

*Pristionchus pacificus* strain PS312 (*Dieterich et al., 2008*) was grown on nematode growth media (NGM) plates supporting a bacterial lawn of either pure *E. coli* OP50 or alternatively a mix of *E. coli* OP50, *Pst* DC3000, and *Lysinibacillus xylanilyticus* (a strain isolated from wild *P. pacificus*). The worms were washed extensively with PBS buffer to remove epidermally-attached bacteria, and DNA was prepared from whole worms using the same bead beating protocol as described for *A. thaliana* (*Regalado et al., 2020*). Worm DNA from the pure culture and the mixed culture were each diluted to 6 ng/µL, and combined in 0:7, 1:6, 2:5, 3:4, 4:3, 5:2, 6:1, and 7:0 ratios to create a titration panel. Each hamPCR template (5 µL template or 30 ng total) was used to perform the HM-tagging reaction, using primers 799F1_G-46628 and 1192R1_G-46629 (V5V6V7 16S rDNA), and Pp_csq-1_F1_G-46691 and Pp_csq-1_R1_G-46692 (*P. pacificus csq-1*). Each exponential PCR reaction was completed in a single reaction of 25 µL; the titration was replicated three times.

## *Triticum aestivum* titration panel

*Triticum aestivum* (wheat) seeds (Rapunzel Naturkost, Legau, Germany) were surface-sterilized by immersion in 70% ethanol and 0.1% Triton X-100 for 1 min, soaking for 15 min in 10% household bleach, and finally washing three times in sterile autoclaved water. Axenic plants were grown on 1% agar supplemented with 1/2 strength MS medium buffered with MES. About 1 g of sterile leaf tissue was harvested after 10 days, and DNA was extracted in the sterile hood as described in ref. (*Regalado et al., 2020*). Roots that had been spontaneously colonized by microbes were obtained by growing by transplanting germinated seeds outdoors into potting soil. Roots were harvested from approximately 4-week-old plants and surface-sterilized by immersion in 10% household bleach with 0.1% Triton X-100 for 5 min, followed by three washes with sterile water. Axenic leaf DNA and spontaneously colonized root DNA were each diluted to 60 ng/µL and combined in 0:7, 1:6, 2:5, 3:4, 4:3, 5:2, 6:1, and 7:0 ratios to create a titration panel of eight samples. Each hamPCR HM-tagging reaction used 3 µL (~180 ng) template; fungal ITS1 HM-tagging was performed with primers ITS1_F1_G-46622 and ITS2_R1_G-46623 (ITS1 rDNA), and PolA1_F1_G-46750 and PolA1_R1_G-46751 (*T. aestivum* RNA polymerase one gene, *PolA1*). Bacterial 16S rDNA HM-tagging was performed with the same *PolA1* primers and with 515_F1_G-46603 and 799_R1_G-46601 (V4 16S

rDNA). To make an additional ITS1 library enriched for ITS1 amplicons, the ITS1 primer pair concentration was increased by a factor of 1.33 and the *PolA1* primer pair concentration was decreased by a factor of 0.66, giving a 2:1 ratio instead of the standard 1:1 ratio, and the tagged products were amplified with 7 HM-tagging and 25 PCR cycles instead of the standard 2 HM-tagging and 30 PCR cycles.

### *Zea mays* field samples

Samples of leaves from mature *Zea mays* (maize) genotype B73 were harvested by standard hole punch from a field side in Tübingen. Permission to punch the leaves was graciously provided by Dr. Marja Timmermans (University of Tübingen). Each sample comprised five leaf discs, which were immediately shaken in 1 mL of sterile water in a screw cap tube to remove dust from the field. The water was removed by pipetting and the leaf discs were snap frozen in liquid nitrogen and taken back to the lab for processing. DNA was extracted with the bead beating protocol described above, with the difference that prior to addition of lysis buffer and garnet rocks, the deep frozen leaf discs were pre-ground with three metal ball bearings at a speed of 5.0 m/s in a FastPrep-24 instrument. We found this pre-grind was helpful to break down the fibrous maize leaf tissue. Prior to adding garnet rocks and lysis buffer, the metal balls were removed by magnet, as metal balls can crack the tubes at the speed of 6.0 m/s used for the primary DNA extraction. Each hamPCR HM-tagging reaction used 10 µL (~120 ng) template; Bacterial 16S rDNA was tagged with one of the forward primers 515F_bcGA_G-47188, 515F_bcTC_G-47189, 515F_bcAG_G-47190, or 515_F3_G-46694 paired with the reverse primer 799_R1_G-46601 (V4 16S rDNA). Maize *LUMINIDEPENDENS (LD)* was tagged with one of the forward *LDP1* primers Zm_LD_bcGA_G-47184, Zm_LD_bcTC_G-47185, Zm_LD_bcAG_G-47186, or Zm_LD_bcCT_G-47187 paired with the *LD* reverse primer Zm_LD_R_G-47158. Two HM-tagging cycles were paired with 30 exponential cycles. To reduce host representation in the final library from the original ~75% to approximately 40%, we used the gel remixing technique described in *Figure 3*.

## Test of HM-tagging step cycle numbers

As templates, we used a pool of mixed wild *A. thaliana* leaf DNA (~ 50 ng / reaction) and the 'synthetic equimolar plasmid template' (~ 0.05 ng / reaction, Appendix 1 - Discussion 2). For the wild *A. thaliana* leaf DNA, we tested V4 16S rDNA primers alone in the HM-tagging step vs. hamPCR with V4 16S rDNA primers plus primers for the host *GI* gene. For the 'synthetic equimolar plasmid template', we used only hamPCR. Specifically, we used 515_F1_G-46603 and 799_R1_G-46601 (V4 16S rDNA) and At.GI_F1_G-46602 and At.GI_R502bp_G-46614 (502 bp *GI* gene).

We applied hamPCR for 2, 3, 4, 5, 6, 7, 8, 9, or 10 HM-tagging cycles, paired with 30, 29, 28, 27, 26, 25, 24, 23, or 22 PCR cycles, respectively. All HM-tagging and PCR reactions were started together, and fewer HM-tagging cycles than 10, or fewer PCR cycles than 30, were achieved by taking PCR tubes out of the thermocycler at the end of the appropriate extension steps and placing them on ice.

## Tests of template concentrations

A panel of eight concentrations of wild *A. thaliana* leaf DNA was prepared, ranging from 5 to 500 ng per reaction. Primers for the wild *A. thaliana* leaf DNA were 515_F1_G-46603 and 799_R1_G-46601 (V4 16S rDNA) and At.GI_F1_G-46602 and At.GI_R502bp_G-46614 (502 bp *GIGANTEA* gene), with both primer pairs in equal ratio.

## Quantitative real-time PCR on *C. annuum* samples

A primer set targeting the gene for the type III effector XopQ of pathogenic *Xanthomonas* was used to measure abundance of *Xe* 85–10 (*Doddaraju et al., 2019*). For *C. annuum*, primers targeting the *UBI-3* gene encoding a ubiquitin-conjugating protein were used (*Wan et al., 2011*). Two reagent mastermixes were prepared, one for each primer set, to help improve primer dose consistency. Each sample was amplified using three 10 µL technical replicates per primer set that were averaged for analysis. Each 10 µL reaction included 2.5 µL of DNA, to which was added, as a mastermix, 5 µL SYBR Green PCR Master Mix (Life Technologies, Carlsbad, California), 1.5 µL water, 0.5 µL of 5 µM forward primer, and 0.5 µL of 5 µM reverse primer. qPCR was performed on a BioRad CFX384 Real-

time System and analyzed with the CFX Manager Software. The following conditions were used for the amplification of both target genes:

1. 94°C for 5 min. *Denature*
2. 94°C for 30 s. *Denature*
3. 55°C for 30 s. *Annealing*
4. 68°C for 45 s. *Extension*
5. Image fluorescence
6. GO TO STEP 2 for 39 additional cycles

The ratio of microbial to host DNA was initially calculated as $2^{(-mean\ xopQ\ Cq\ value)} / 2^{(-mean\ UBI-3\ Cq\ value)}$. See alignment to CFU counts below.

## Alignment of *Xanthomonas* qPCR and hamPCR load with CFU counts

Log10-transformed *Xe* 85–10 ASV loads from hamPCR were regressed onto log10-transformed *xopQ* loads from qPCR (least squares method), and the slope ($m$) and y-intercept ($b$) of the best-fit line were used to transform and align the qPCR loads to hamPCR loads with the following formula: Load_qPCR$_{hamPCR-scaled}$ = $m$ × Load_qPCR + $b$. Next, log10-transformed CFU counts were regressed onto the log10-transformed hamPCR loads, and the slope and y-intercept of the resulting best-fit line were used similarly to align both Load_qPCR$_{hamPCR-scaled}$ and hamPCR loads to the CFU counts.

## Correlation networks

Pearson correlation matrices for relative abundance and microbial load data were created in R (*R Development Core Team, 2019*) using the 'stats' package. The SparCC (*Friedman and Alm, 2012*) correlation matrix was created in R using the implementation in the 'SpiecEasi' package (*Kurtz et al., 2020*). Networks were visualized with the package 'qgraph' (*Epskamp et al., 2012*). Custom scripts are available on GitHub (https://github.com/derekLS1/hamPCR).

# Acknowledgements

We thank Lei Li and Fiona Beitel for discussions and suggestions regarding PCR and qPCR, Talia Karasov for suggestions on presentation, Katrin Fritschi and Heike Budde for technical help with sequencing, Ilja Bezrukov for assistance with demultiplexing, and Dor Russ, Jeff Dangl, and Benjamin Schwessinger for supportive discussions and comments on the manuscript. Thanks to Victor Schmidt, Nick Youngblut, Lei Li, and Hua Wang for giving us space in their HiSeq 3000 lanes. Thanks to Marja Timmermans for access to field maize plants.

# Additional information

## Competing interests

Detlef Weigel: Senior editor, *eLife*. The other authors declare that no competing interests exist.

## Funding

| Funder | Grant reference number | Author |
| --- | --- | --- |
| Deutsche Forschungsgemeinschaft | SPP 2125 DECRyPT | Derek S Lundberg<br>Pratchaya Pramoj Na Ayutthaya<br>Detlef Weigel |
| Human Frontier Science Program | LT000565/2015-L | Derek S Lundberg |
| Max Planck Society | | Derek S Lundberg<br>Pratchaya Pramoj Na Ayutthaya<br>Gautam Shirsekar<br>Wen-Sui Lo<br>Detlef Weigel |
| Deutsche Forschungsge- | 390838134 | Derek S Lundberg |

meinschaft

Pratchaya Pramoj Na Ayutthaya
Gautam Shirsekar
Wen-Sui Lo
Detlef Weigel

The funders had no role in study design, data collection and interpretation, or the decision to submit the work for publication.

## Author contributions

Derek S Lundberg, Conceptualization, Data curation, Software, Formal analysis, Supervision, Funding acquisition, Validation, Investigation, Visualization, Methodology, Writing - original draft, Project administration, Writing - review and editing; Pratchaya Pramoj Na Ayutthaya, Conceptualization, Formal analysis, Validation, Investigation, Visualization, Methodology, Writing - original draft, Writing - review and editing; Annett Strauß, Gautam Shirsekar, Conceptualization, Resources, Investigation, Methodology, Writing - review and editing; Wen-Sui Lo, Resources, Methodology, Writing - review and editing; Thomas Lahaye, Conceptualization, Methodology, Writing - review and editing; Detlef Weigel, Conceptualization, Resources, Funding acquisition, Validation, Methodology, Writing - original draft, Writing - review and editing

## Author ORCIDs

Derek S Lundberg (iD) https://orcid.org/0000-0001-7970-1595
Detlef Weigel (iD) https://orcid.org/0000-0002-2114-7963

## Decision letter and Author response

Decision letter https://doi.org/10.7554/eLife.66186.sa1
Author response https://doi.org/10.7554/eLife.66186.sa2

## Additional files

### Supplementary files

• Supplementary file 1. Primers, metadata, and other raw data. This file is an excel sheet with six subtables as tabs. These are: 'Tagging Primers', 'PCR Primers', 'qPCR Primers', 'Sample Metadata', 'Pepper qPCR raw data', and 'Pepper CFU raw data'.

• Transparent reporting form

### Data availability

All data in this manuscript have been deposited in the European Nucleotide Archive (ENA) under the project number PRJEB38287.

The following dataset was generated:

| Author(s) | Year | Dataset title | Dataset URL | Database and Identifier |
|---|---|---|---|---|
| Lundberg DS, Pramoj P, Strauß A, Shirsekar G, Lo W-S, Lahaye T, Weigel D | 2020 | hamPCR | https://www.ebi.ac.uk/ena/browser/view/PRJEB38287 | European Nucleotide Archive, PRJEB38287 |

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

## Appendix 1

### Primers: Microbe tagging

Standard desalting is sufficient purification for the HM-tagging primers.

### V4 region (16S rDNA), 515F - 799 R / 515F - 806R, ~414 bp final amplicon

Underlined letters represent the universal overhang.

799F is commonly used in combination with 1192R because it avoids plant plastids by mismatch. Shown here is the reverse complement of that primer. Used in combination with PNAs (*Lundberg et al., 2013*), it avoids nearly all plastids.

Bracketed letters represent 'linkers' - sequences designed to anneal to as few 16S rDNA molecules as possible, and ensure that only the gene-specific region binds to the template.

Lowercase letters are template-specific. Template-specific primer sequences for 515F and 806R are the same as used in the Earth Microbiome Project *Thompson et al., 2017* and *Lundberg et al., 2013*. Linker sequences for 515F and 806R are as in *Lundberg et al., 2013*. Template-specific primer sequence for 799F as in *Agler et al., 2016b*.

'F1', 'F3', 'bcGA', 'bcTC', and 'bcAG' forward primers perform identically but each introduces different sequenced bases which serve as an additional barcode. See *Lundberg et al., 2013*.

FORWARD:

```
>515_F1_G-46603
TCCCTACACGACGCTCTTCCGATCT [GA] gtgycagcmgccgcggtaa
>515_F3_G-46694
TCCCTACACGACGCTCTTCCGATCT CT [GA] gtgycagcmgccgcggtaa
>515F_bcGA_G-47188
TCCCTACACGACGCTCTTCCGATCT GA [GA] gtgycagcmgccgcggtaa
>515F_bcTC_G-47189
TCCCTACACGACGCTCTTCCGATCT TC [GA] gtgycagcmgccgcggtaa
>515F_bcAG_G-47190
TCCCTACACGACGCTCTTCCGATCT AG [GA] gtgycagcmgccgcggtaa
```

REVERSE 799:

```
>799_R1_G-46601
GGAGTTCAGACGTGTGCTCTTCCGATCT [TG] cmgggtatctaatcckgtt
```

REVERSE 806:

```
>806_R1_G-46631
GGAGTTCAGACGTGTGCTCTTCCGATCT  [AC] ggactacnvgggtwtctaat

>E. coli_V4_sequenced_region_285bp_final_amplicon_413bp
gtgccagcagccgcggtaaTACGGAGGGTGCAAGCGTTAATCGGAATTACTGGGCGTAAAGCGCACGCAGG
CGGTTTGTTAAGTCAGATGTGAAATCCCCGGGCTCAACCTGGGAACTGCATCTGATACTGGCAAGCTTGAG
TCTCGTAGAGGGGGGTAGAATTCCAGGTGTAGCGGTGAAATGCGTAGAGATCTGGAGGAATACCGGTGGCG
AAGGCGGCCCCCTGGACGAAGACTGACGCTCAGGTGCGAAAGCGTGGGGAGCAaacaggattagataccct
g
```

### V3V4 region (16S rDNA), 341F - 799R, ~587 bp final amplicon

Underlined letters represent the universal overhang.

799F is commonly used in combination with 1192R because it avoids plant plastids by mismatch. Shown here is the reverse complement of that primer. Used in combination with PNAs (*Lundberg et al., 2013*), it avoids nearly all plastids.

Bracketed letters represent 'linkers' - sequences designed to anneal to as few 16S rDNA molecules as possible, and ensure that only the gene-specific region binds to the template.

Lowercase letters are template-specific. Template-specific primer sequence for 341F is as in *Agler et al., 2016b*.

## FORWARD:

```
>341_F1_G-46605
TCCCTACACGACGCTCTTCCGATCT [GA] cctacgggaggcagcag
```

## REVERSE:

```
>799_R1_G-46601
GGAGTTCAGACGTGTGCTCTTCCGATCT [TG] cmgggtatctaatcckgtt
```

```
> E. coli _V3V4_sequenced_region_459bp_final_amplicon_587bp
cctacgggaggcagcagTGGGGAATATTGCACAATGGGCGCAAGCCTGATGCAGCCATGCNGCGTGTATGA
AGAAGGCCTTCGGGTTGTAAAGTACTTTCAGCGGGGAGGAAGGGAGTAAAGTTAATACCTTTGCTCATTGA
CGTTACCCGCAGAAGAAGCACCGGCTAACTCCGTGCCAGCAGCCGCGGTAATACGGAGGGTGCAAGCGTTA
ATCGGAATTACTGGGCGTAAAGCGCACGCAGGCGGTTTGTTAAGTCAGATGTGAAATCCCCGGGCTCAACC
TGGGAACTGCATCTGATACTGGCAAGCTTGAGTCTCGTAGAGGGGGGTAGAATTCCAGGTGTAGCGGTGAA
ATGCGTAGAGATCTGGAGGAATACCGGTGGCGAAGGCGGCCCCCTGGACGAAGACTGACGCTCAGGTGCGA
AAGCGTGGGGAGCAaacaggattagatacccctg
```

## V5V6V7 (16S rDNA) region, 799F - 1192R, ~543 bp final amplicon

Underlined letters represent the universal overhang.

Bracketed letters represent 'linkers' - sequences designed to anneal to as few 16S rDNA molecules as possible, and ensure that only the gene-specific region binds to the template.

Lowercase letters are template-specific. Template-specific primer sequences are as in *Agler et al., 2016b*.

## FORWARD:

```
>799_F1_G-46628
TCCCTACACGACGCTCTTCCGATCT [GT] aacmggattagataccckg
```

## REVERSE:

```
>1192_R1_G-46629
GGAGTTCAGACGTGTGCTCTTCCGATCT [GT] acgtcatccccaccttcc
```

```
>E. coli_V5V6V7_sequenced_region_414bp_final_amplicon_543bp
aacaggattagatacccctgGTAGTCCACGCCGTAAACGATGTCGACTTGGAGGTTGTGCCCTTGAGGCGTG
GCTTCCGGAGCTAACGCGTTAAGTCGACCGCCTGGGGAGTACGGCCGCAAGGTTAAAACTCAAATGAATTG
ACGGGGGCCCGCACAAGCGGTGGAGCATGTGGTTTAATTCGATGCAACGCGAAGAACCTTACCTGGTCTTG
ACATCCACGGAAGTTTTCAGAGATGAGAATGTGCCTTCGGGAACCGTGAGACAGGTGCTGCATGGCTGTCG
TCAGCTCGTGTTGTGAAATGTTGGGTTAAGTCCCGCAACGAGCGCAACCCTTATCCTTTGTTGCCAGCGGT
CCGGCCGGGAACTCAAAGGAGACTGCCAGTGATAAACTGGAggaaggtggggatgacgt
```

## ITS oomycete, ITS1-O_F - 5.8s-O_R, ~394 bp final amplicon

Underlined letters represent the universal overhang.

Lowercase letters are template-specific. Template-specific primer sequences are as in *Agler et al., 2016a*.

### FORWARD:

```
>ITS1-O_F1_G-46636
TCCCTACACGACGCTCTTCCGATCT cggaaggatcattaccac
```

### REVERSE:

```
>5.8s-O_R1_G-46637
GGAGTTCAGACGTGTGCTCTTCCGATCT agcctagacatccactgctg

>ITS_H.arabidopsidis_sequenced_region_269bp_final_amplicon_394bp
cggaaggatcattaccacACCTAAAAAACTTTCCACGTGAACCGTTTCAACCCAATAGTTGGGGGTCTTAT
TTGGCGGCGGCTGCTGGCTTAATTGTTGGCGGCTGCTGCTGAGTGAGCCCTATCAAAAAAAAGGCGAACGT
TTGGGCTTCGGCCTGATTTAGTAGTCTTTTTTTCTTTTAAACCCCTTCCTTAATACTGAATATACTGTGGG
GACGAAAGTCTCTGCTTTTAACTAGATAGCAACTTTcagcagtggatgtctaggct
```

## ITS1 fungus, ITS1F - ITS2, variable length final amplicon

Underlined letters represent the universal overhang.

Bracketed letters represent 'linkers' - sequences designed to anneal to as few ITS molecules as possible, and ensure that only the gene-specific region binds to the template.

Lowercase letters are template-specific. Template-specific primer sequences and linker sequences are as used in the Earth Microbiome Project (*Thompson et al., 2017*).

### FORWARD:

```
>ITS1_F1_G-46622
TCCCTACACGACGCTCTTCCGATCT [GG] cttggtcatttagaggaagtaa
```

### REVERSE:

```
>ITS2_R1_G-46623
GGAGTTCAGACGTGTGCTCTTCCGATCT [CG] gctgcgttcttcatcgatgc

>ITS1_Agaricus_bisporus_sequenced_region_407bp_final_amplicon_532bp
cttggtcatttagaggaagtaaAAGTCGTAACAAGGTTTCCGTAGGTGAACCTGCGGAAGGATCATTATTG
AATTATGTTTTCTAGATGGGTTGTAGCTGGCTCTTCGGAGTATGTGCACGCCTGTCTGGACTTCATTTTCA
TCCACCTGTGCACCTTTTGTAGTCTTTTTCAGGTATTGGAGGAAGTGGTCAGCCTATCAGCTCTTTGCTGG
ATGTAAGGACTTGCAGTGTGAAAACAGTGCTGTCCTTTACCTTGGCCATGGAATCTTTTTCCTGTTAGAGT
CTATGTTATTCATTATACTCTTAGAATGTCATTGAATGTCTTTACATGGGCTATGCCTATGAAAATTATTA
TACAACTTTCAGCAACGGATCTCTTGGCTCTCgcatcgatgaagaacgcagc
```

## Actin gene from *Hyaloperonospora arabidopsidis*, 585 bp final amplicon

Underlined letters represent the universal overhang.

Lowercase letters are template-specific. The template-specific primer sequences were chosen for this study; the choice of the Actin gene was based on (*Anderson and McDowell, 2015*).

```
>Ha.Actin_F1_G-46716
TCCCTACACGACGCTCTTCCGATCT cgcgctgccgcacgcgattgt

>Ha.Actin_R1_G-46717
GGAGTTCAGACGTGTGCTCTTCCGATCT cgccaaagccgtcagttccttc

>Actin_Hyaloperonospora_arabidopsidis_Emoy2_
sequenced_region_457bp_final_amplcion_585bp
cgcgctgccgcacgcgattgtGCGTTTGGATCTCGCTGGTCGCGACTTGACCGACTACATGATGAAGATCT
TGACGGAGCGCGGGTACTCGTTTACTACCACGGCCGAGCGCGAAATCGTGCGCGACATTAAAGAGAAACTC
ACGTACATTGCACTGGACTTTGACCAGGAGATGAAGACAGCGGCCGAGTCGTCGGGACTCGAGAAGAGCTA
CGAATTGCCGGATGGCAATGTGATTGTCATTGGCAATGAACGTTTCCGTACGCCGGAAGTGCTGTTCCAGC
CGTCGCTCATTGGCAAAGAAGCTGCCGGTATTCACGACTGCACGTTCCAGACCATCATGAAGTGTGACGTG
GATATCCGGAAGGATTTGTACTGCAACATTGTGCTCTCGGGCGGAACCACCATGTACCCGGGCATTGGCGA
ACGCATGACgaaggaactgacggctttggcg
```

## Primers: Host tagging

*GIGANTEA* (*GI*) from *Arabidopsis thaliana,* 502 bp final amplicon

Size distinguishable from V4 or V3V4 16S rDNA amplicons

Underlined letters represent the universal overhang.

Lowercase letters are template-specific. Here, the template-specific portion extends into the universal overhang for the At.GI_F1_G-46602 primer.

'F1' and 'F3' forward primers perform identically but F3 introduces a 'CT' causing a frameshift that can also serve as an additional barcode. See (*Lundberg et al., 2013*). Here, the additional 'CT' extends into the universal overhang for the At.GI_F3_G-46640 primer.

FORWARD:

```
>At.GI_F1_G-46602
TCCCTACACGACGCTCTTCCGATct gtaaagataaatgggtcatctaa
>At.GI_F3_G-46640
TCCCTACACGACGCTCTTCCGATCT ctgtaaagataaatgggtcatctaa
>At.GI_F_bcAG_G-46852
TCCCTACACGACGCTCTTCCGATCT AG ctgtaaagataaatgggtcatctaa
>At.GI_F_bcTA_G-46853
TCCCTACACGACGCTCTTCCGATCT TA ctgtaaagataaatgggtcatctaa
>At.GI_F_bcCT_G-46854
TCCCTACACGACGCTCTTCCGATCT CT ctgtaaagataaatgggtcatctaa
```

REVERSE:

```
>At.GI_R502bp_G-46614
GGAGTTCAGACGTGTGCTCTTCCGATCT tccttctgaaccggtgtattc

>Athaliana_GI_sequenced_region_377bp_final_amplicon_502bp
gtaaagataaatgggtcatctaaAGAGTATGGAGCTGGGATTGACTCGGCAATTAGTCATACGCGCCGAAT
TTTGGCAATCCTAGAGGCACTCTTTTCATTAAAACCATCTTCTGTGGGGACTCCATGGAGTTACAGTTCTA
GTGAGATAGTTGCTGCGGCCATGGTTGCAGCTCATATTTCCGAACTGTTCAGACGTTCAAAGGCCTTGACG
CATGCATTGTCTGGGTTGATGAGATGTAAGTGGGATAAGGAAATTCATAAAAGAGCATCATCATTATATAA
CCTCATAGATGTTCACAGCAAAGTTGTTGCCTCCATTGTTGACAAAGCTGAACCCTTGGAAGCCTACCTTA
AgaatacaccggttcagaaggA
```

*GIGANTEA* (*GI*) from *Arabidopsis thaliana*, 466 bp final amplicon

Size distinguishable from V5V6V7 16S rDNA amplicons

Underlined letters represent the universal overhang.

Lowercase letters are template-specific. Here, the template-specific portion extends into the universal overhang for the At.GI_F1_G-46602 primer.

FORWARD:

```
>At.GI_F1_G-46602
TCCCTACACGACGCTCTTCCGATct gtaaagataaatgggtcatctaa
```

REVERSE:

```
>At.GI_R466bp_G-46652
GGAGTTCAGACGTGTGCTCTTCCGATCT aagggttcagctttgtcaacaa

>A.thaliana_GI_sequenced_region_341bp_final_amplicon_466bp
gtaaagataaatgggtcatctaaAGAGTATGGAGCTGGGATTGACTCGGCAATTAGTCATACGCGCCGAAT
TTTGGCAATCCTAGAGGCACTCTTTTCATTAAAACCATCTTCTGTGGGGACTCCATGGAGTTACAGTTCTA
GTGAGATAGTTGCTGCGGCCATGGTTGCAGCTCATATTTCCGAACTGTTCAGACGTTCAAAGGCCTTGACG
CATGCATTGTCTGGGTTGATGAGATGTAAGTGGGATAAGGAAATTCATAAAAGAGCATCATCATTATATAA
CCTCATAGATGTTCACAGCAAAGTTGTTGCCTCCAttgttgacaaagctgaaccctt
```

*GIGANTEA* (*GI*) from *Capsicum annuum*, 508 bp final amplicon

Size distinguishable from V4 or V3V4 16S rDNA amplicons

Underlined letters represent the universal overhang.

Lowercase letters are template-specific.

FORWARD:

```
>Ca.GI_F1_G-46626
TCCCTACACGACGCTCTTCCGATCT caaatgattcatctattgagcta
```

REVERSE:

```
>Ca.GI_R1_G-46627
GGAGTTCAGACGTGTGCTCTTCCGATCT ctcctttagaactggtgcatg

>C.annuum_GI_sequenced_region_371bp_final_amplicon_508bp
caaatgattcatctattgagctaCGAAATGGGATTCATTCTGCCGTCTCTCATACTCGGAGGATATTGGCA
ATTTTAGAGGCACTTTTTTCTCTGAAACCATCGTCTGTTGGAACCTCATGGAGCTACAGCTCAAATGAGAT
AGTTGCTGCAGCTATGGTAGCTGCTCACATTTCTGATCTGTTTAGACACAACAAGGCCTGCATGCAAGCTC
TTTCTATTTTGATACGGTGTAAGTGGGATAATGAAATTCATTCCAGGGCATCGTCACTTTATAACCTAATT
GATATTCATAGCAAAACTGTTGCATCAATTGTCAACAAGGCTGAACCATTGGAAGCTTATCTAATAcatgc
accagttctaaaggag
```

RNA polymerase I gene (*PolA1*) from *Triticum aestivum*, 497 bp final amplicon

Size distinguishable from V4 or V3V4 16S rDNA amplicons

Underlined letters represent the universal overhang.

Lowercase letters are template-specific. Here, the template-specific portion extends into the universal overhang for both primers.

FORWARD:

```
>Ta.PolA1_F1_G-46750
TCCCTACACGACGCTCTTCCGATct gatgttgtggaaggaattgaa
```

REVERSE:

```
>Ta.PolA1_R1_G-46751
GGAGTTCAGACGTGTGCTCTTCCGAtct gcatcagctcccaagtc
```

```
>T.aestivum_PolA1_sequenced_region_370bp_final_amplicon_497bp
ctgatgttgtggaaggaattgaaGTATGCACAGTTCCTTTTCACAACAGTAATGGGCATATTTCGAGTCTC
TATAAGTTGCATCTAAAACTATTCTCACCCGACTGTTACCCTCCTGAGTCGGAACTTACAGTAGATGAGTG
TCAAGCATCCTTGAGAACTGTGTTTGTTGATGCAATGGAATATGCAATCGAAAAACACCTAAATTTGCTAC
ACAAAGTTAGTGGAATCCAGGAAACAAGGGTAAAGGACACTGAGAGTTTACCATCAGAAGGTCCTGAAGAA
TCCGAGGGCAGACCTACCAACGGGGACGAGTCTGATACGAGTGATGGTGACGATGAAAATGAGGATgactt
gggagctgatgcaga
```

## Calsequestrin gene (*csq-1*) from *Pristionchus pacificus*, 470 bp final amplicon
Size distinguishable from V5V6V7 16S rDNA amplicons

Underlined letters represent the universal overhang.
  Lowercase letters are template-specific.

FORWARD:

```
>Pp.csq-1_F1_G-46691
TCCCTACACGACGCTCTTCCGATCT catcgagaatatatggccgat
```

REVERSE:

```
>Pp.csq-1_R1_G-46692
GGAGTTCAGACGTGTGCTCTTCCGATCT tgagttatgccccactattcaa
```

```
>P.pacificus_csq-1_sequenced_region_343bp_final_amplicon_470bp
catcgagaatatatggccgatCGAGAATGTTCCGATTTGCCAGTGGAGAGATGTACAGGATGGAATGTCAC
ATGCAGATGTGCTTGAAGGCATCGACTTGCGGGCCGAGGACTACTGTAAGTTACATAGAAACCAAAGATTG
ACTTTCTAGTTGGTTCAAAATTTTCTCGAAAAGGATTTGAGAATATATAACAATTGAGAACTATTCTAATT
GAGTAGCAGTTAGACTATAATTACCGAATATTCTTTGAATGCTCACTAATACTCAATTTCCCCTGTCGTTA
TGCTCGTAATTCACCATGCAGTACCTCTTCCTGTCTGttgaatagtggggcataactca
```

## *LUMINIDEPENDENS* (*LD*) from *Zea mays*, 470 bp final amplicon
Size distinguishable from V4 or V3V4 16S rDNA amplicons

Underlined letters represent the universal overhang.
  Bracketed letters represent 'linkers' - sequences designed not to anneal to the template and reduce unintended binding of the rest of the primer.
  Lowercase letters are template-specific.
  'F1', 'bcGA', 'bcTC', 'bcAG', and 'bcCT' forward primers perform identically but each introduces different sequenced bases which serve as an additional barcode. See *Lundberg et al., 2013*.

FORWARD:

```
>Zm_LD_bcGA_G-47184
TCCCTACACGACGCTCTTCCGATCT GA [TA] tctgccctggtggagtacccat
>Zm_LD_bcTC_G-47185
TCCCTACACGACGCTCTTCCGATCT TC [TA] tctgccctggtggagtacccat
>Zm_LD_bcAG_G-47186
TCCCTACACGACGCTCTTCCGATCT AG [TA] tctgccctggtggagtacccat
>Zm_LD_bcCT_G-47187
TCCCTACACGACGCTCTTCCGATCT CT [TA] tctgccctggtggagtacccat
```

REVERSE:

```
>Zm.LD_R1_G-47158
GGAGTTCAGACGTGTGCTCTTCCGATCT cctcggcgcctccgggtt

>Z.mays_LD_sequenced_region_375bp_final_amplicon_512bp
gccctggtggagtacccatGGGACGGGCACCACCAGCGGCACTCGAGGTCACCAGATCCTGGCGTGGTCCG
GGACTACGACACCGACTATGGCGGTGCGCAGGGCTACAGTCAGCAGCCCCTGACGCAGTGGAGTGCAGGGA
AAGTGCAGCAGCAGGGCTACAATCCTGAGCCGTCAAGGCAGTGGAGTTCTTCCCAGGCGCACCAGGGCGGC
TACGCACCCGCCGAGCCGTCGAGGCAGTGGAGTTCTTCCCAGGCGCACCAGAGCTACGCTCCCGAGCTACC
GAGGCAGTGGAGCTCCGAACGCCGTGGCTACGATGATGCGGAGCCCTCGAGGACATGGAGCTCCGGCCAGC
AGaacccggaggcgccgagg
```

## Discussion 1: Note on Adapting HM-tagging Sets for Other Protocols

Simply changing the universal overhangs and adjusting annealing temperatures accordingly may be sufficient to use hamPCR with other two-step PCR protocols. *The following are starting suggestions only, and have not been tested experimentally.*

To adapt a forward HM-tagging primer to that in *Gohl et al., 2016*, replace the *upper-case, unbracketed letters* in the forward primer for that template (shown in orange in the example below) with 'TCGTCGGCAGCGTCAGATGTGTATAAGAGACAG'.

For example, for '**At.GI_F1_G-46602**' for *A. thaliana GI*, 'TCCCTACACGACGCTCTTCCGATct gtaaagataaatgggtcatctaa' would become: 'TCGTCGGCAGCGTCAGATGTGTATAAGAGACAG ctgtaaagataaatgggtcatctaa'

To adapt a reverse HM-tagging primer to that in *Gohl et al., 2016*, replace the *upper-case, unbracketed letters* (shown in orange in the example below) in the reverse primer for that template with: 'GTCTCGTGGGCTCGGAGATGTGTATAAGAGACAG'.

For example, for '**At.GI_R502bp_G-46614**' for *A. thaliana GI*, 'GGAGTTCAGACGTGTGCTCTTCCGATCT tccttctgaaccggtgtattc' (this work) would become: 'GTCTCGTGGGCTCGGAGATGTGTATAAGAGACAG tccttctgaaccggtgtattc'.

Rather than adapt the forward HM-tagging primers to work with the forward PCR primer in *Lundberg et al., 2013*, we recommend using the forward primer here. The reverse HM-tagging primers can be used directly with the reverse PCR primers in *Lundberg et al., 2013*.

## Primers: Exponential PCR

For the full PCR primer set, please see *Supplementary file 1*. PCR primers should be HPLC-purified or purified with an alternative method to reduce truncated primers.

FORWARD

```
>PCR_F_G-40610
AATGATACGGCGACCACCGAGATCTACACTCTTTCCCTACACGACGCTCTTC
```

REVERSE:

Example PCR_R_indexed primers. Same set as that published in *Lundberg et al., 2013*.

| Model Reverse Primer | **CAAGCAGAAGACGGCATACGAGAT XXXXXXXXX GTGACTGGAGTTCAGACGTG TGCTC** |
|---|---|
| PCR_R_bc1 | CAAGCAGAAGACGGCATACGAGAT TTACCGACG GTGACTGGAGTTCAGACGTGTGCTC |
| PCR_R_bc2 | CAAGCAGAAGACGGCATACGAGAT ATTGGACAC GTGACTGGAGTTCAGACGTGTGCTC |
| PCR_R_bc3 | CAAGCAGAAGACGGCATACGAGAT TCGCATGGA GTGACTGGAGTTCAGACGTGTGCTC |
| Etc. | |

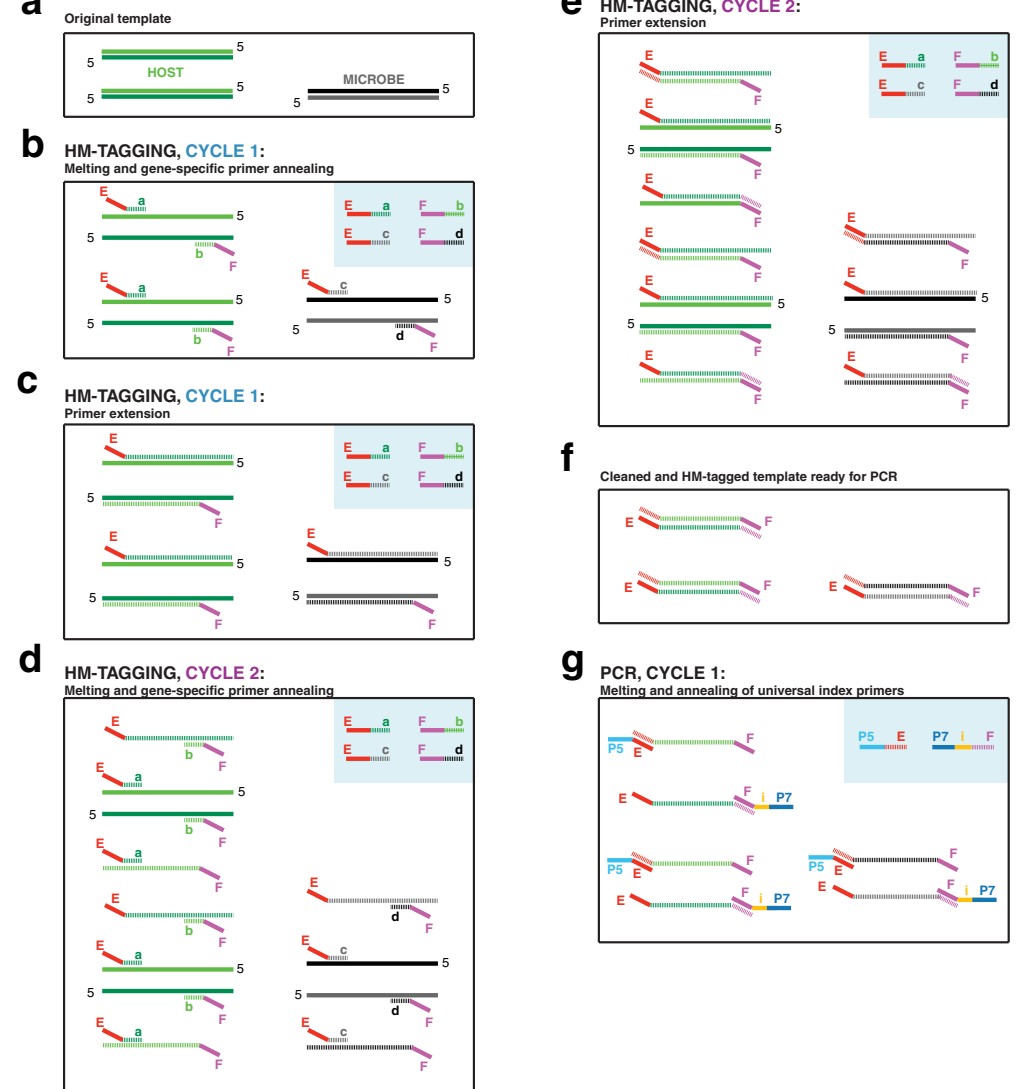

**Appendix 1—figure 1.** PCR Scheme. (**a**) The input DNA is shown as two double-stranded DNA molecules from the host (light/dark green) and one double-stranded DNA molecule from a microbe (black/gray). (**b**) In the annealing step of HM-tagging cycle one, an excess of two primer pairs is present (blue box), each with a gene-specific region (regions a through d) and universal overhangs (regions E and F). The gene-specific regions of each primer anneal to the templates. (**c**) In the

*Appendix 1—figure 1 continued on next page*

*Appendix 1—figure 1 continued*

extension step of HM-tagging cycle 1, primers are extended to make a single copy of each template molecule with a universal overhang on the 5' end, producing 'single-tagged templates'. Extended molecules are represented by dashed lines, while original templates retain a solid line. (**d**) In HM-tagging cycle 2, again the gene-specific regions of each primer anneal to all template molecules, both to the originals and single-tagged templates. (**e**) In the extension step, note that for those primers that had annealed to single-tagged templates, extension generates the reverse complement of the universal overhang, producing 'double-tagged templates'. (**f**) Primers are removed from the reaction with SPRI beads prior to PCR. Note that the quantity of double-tagged template molecules is the same as the quantity of original template molecules. Although original template and single-tagged templates survive SPRI cleanup and are also present in this step, these lack the universal overhangs and cannot be amplified in PCR and are not shown. (**g**) For the exponential PCR step, an excess of a single primer pair is present (blue box), each with a region complementary to the universal overhangs (regions E and F) and sequencing adapters (regions P5 and P7). An index for multiplexing is present on one or both primers (region i). Because double-tagged templates from both host and microbe have the same universal overhangs, they are not differentially amplified during PCR.

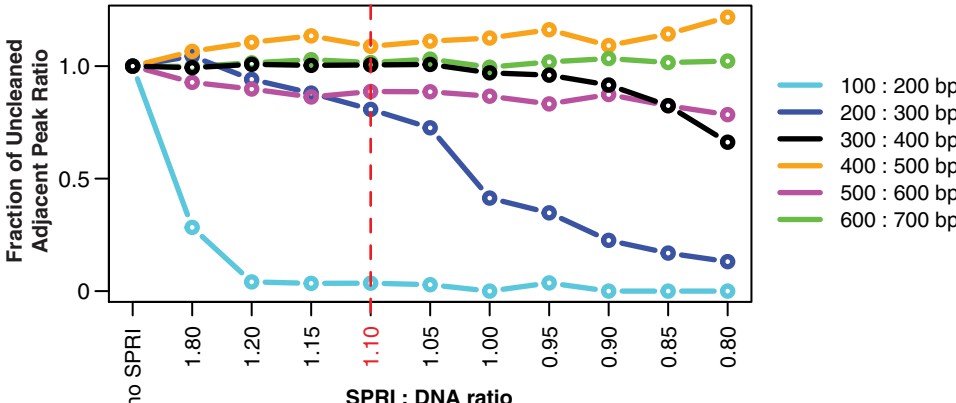

**Appendix 1—figure 2.** SPRI ratios. SPRI beads in a polyethylene glycol (PEG) solution, such as AMPure XP, preferentially bind longer DNA fragments, making them useful for removing free primers and primer dimers from reactions. As the PEG concentration decreases, a wider range of short fragments can no longer bind to the beads, and primer dimers are more completely eliminated. However, if the PEG concentration is too low, the range of fragment sizes eliminated could include DNA of interest. For hamPCR, it is important that the SPRI cleanup does not affect the ratio of the host and microbial amplicons, which could lead to systematic bias and noise. To determine an acceptable PEG concentration, we tested cleanups with SPRI (AMPure XP) beads at different SPRI: DNA ratios (resulting in a range of PEG concentrations) on a standard DNA size ladder (GeneRuler DNA Ladder Mix, Thermo Scientific, Waltham, MA, USA), and quantified abundance of the purified fragments with a Bioanalyzer (Agilent, Santa Clara, CA, USA). Using the pure, uncleaned ladder, we calculated the ratio of each peak's abundance to the adjacent larger peak (200: 300 bp, 300: 400 bp, etc.), and used this set of abundance ratios as a baseline (no SPRI, far left). With each successive decrease in SPRI: DNA ratio (from left to right), we looked for a decrease in the abundance ratio between adjacent bands, which would indicate elimination of the smaller fragment. Of highest interest is the 300: 400 bp ratio (black), because the smallest tagged templates in hamPCR are around 300 bp. We determined SPRI: DNA ratios less than 1.0 endangered the 300: 400 ratio, and thus decided to conservatively use a SPRI: DNA solution of 1.1: 1 or higher for the cleanup of tagged products.

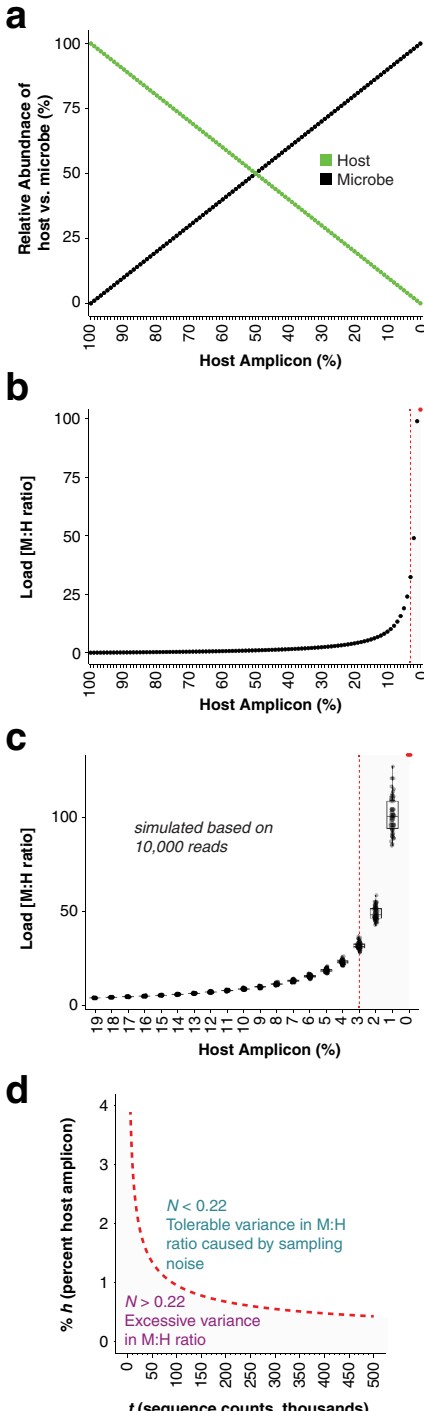

**Appendix 1—figure 3.** In silico load simulations. (**a**), One hundred and one samples were simulated by combining 0 to 100 microbial sequence counts (black points) with 100 to 0 sequence counts (green points). The x-axis shows the percentage of the sample occupied by the host amplicon, decreasing from left to right. The lines show a simple linear, mutually-exclusive relationship between host and microbe. (**b**) The microbial abundances from (**a**) were converted to microbial load by dividing by the host abundances. Note that as the host amplicon abundance approaches 0, the microbial load climbs towards infinity. The red point at 0% host amplicon abundance represents infinite load. The vertical red dotted line indicates 3% host amplicon abundance; below 3% host

*Appendix 1—figure 3 continued on next page*

*Appendix 1—figure 3 continued*

abundance, load becomes extremely sensitive to small changes in host abundance. (**c**) The 100 plant and microbial sequence counts for each sample in (**a**) were multiplied by 10,000 to make virtual samples with 1 million sequence counts. These were then subsampled 50 times each to 10,000 reads to simulate samples with random sampling noise. Microbial load from the virtual samples was calculated as in (**b**) by dividing microbial counts by host counts. Only host amplicon percentages from 19% to 0% are shown to focus on lower host abundances. The red point at 0% host amplicon abundance represents infinite load. The vertical red dotted line indicates the position of 3% host amplicon abundance. Note that not only does load climb quickly towards infinity as host counts approach 0, but also sampling noise has a greater impact on microbial load. (**d**) Deeper sequencing reduces the variance in microbial load associated with a low abundance host amplicon by reducing the impact of sampling noise. We defined a noise level, $N$, as the range in calculated microbial load that would result from subtracting one sequence count from the host and assigning it to a microbe, and vice versa, as shown in the following equation, where $M$ and $H$ are integer sequence counts of microbe and host, respectively: $N = \frac{(M+1)}{(H-1)} - \frac{(M-1)}{(H+1)}$. For 10,000 reads with a host percentage of 3% (300 counts) as shown by the red line in (**c**), we calculate $N = 0.22$ and suggest it as an upper limit. To visualize how deeper sequencing enables lower host amplicon abundances without increasing noise, we plotted the relative abundance of the host amplicon as a function of sequencing depth (red dotted line), maintaining $N = 0.22$ as shown in the following equation, where $t$ is the total reads in each sample and %$h$ is the percent of host amplicon. $\%h = \frac{\sqrt{N(-2t-N)}}{Nt}$. Combinations of host amplicon abundances and sequencing depths below this line (gray area) are less reliable, and not recommended for quantitative conclusions.

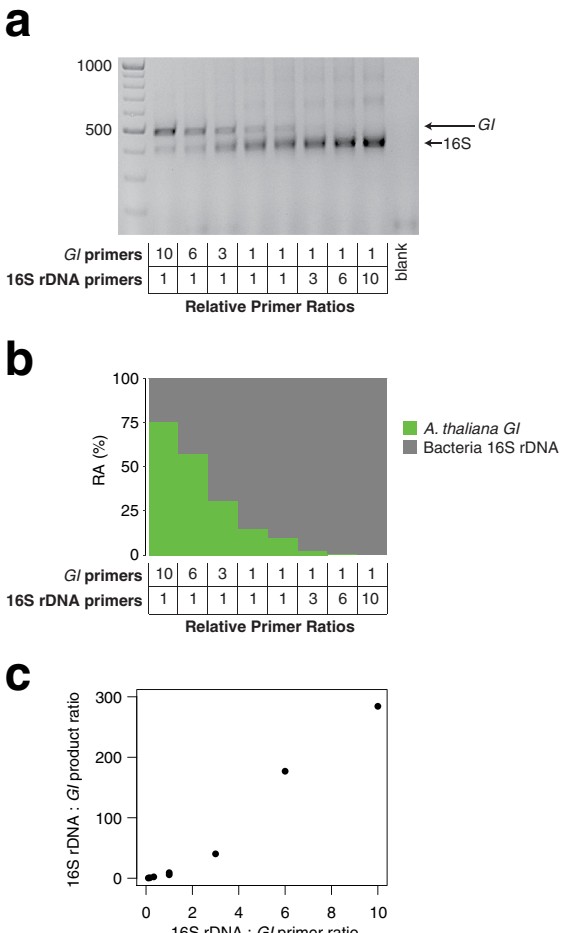

**Appendix 1—figure 4.** Effect of primer pair concentration on product. The same wild *A. thaliana* DNA pool was amplified with hamPCR using eight different ratios of the 16S rDNA primer pair to the *GI* primer pair, ranging from 1/10 to 10. (**a**) 2% agarose gel of the prepared libraries (**b**) Relative abundance (RA) of bacteria and host amplicons in the sequence data. (**c**) Scatterplot of microbe-to-host product ratios plotted against the microbe-to-host primer ratios used to produce them. A tenfold difference in primer ratios resulted in an over 200-fold difference in product ratios, underscoring the importance of steady primer ratios (through the use of a mastermix of primers) across an experiment.

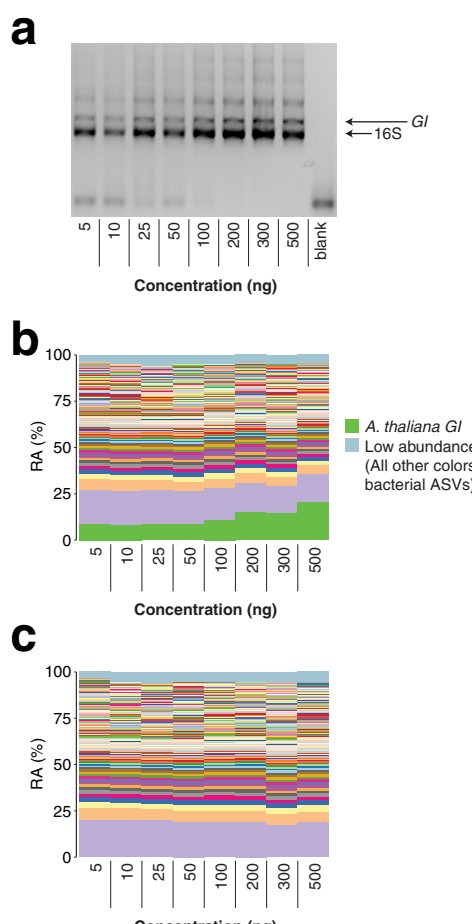

**Appendix 1—figure 5.** Total template concentration test. A panel of eight concentrations of wild *A. thaliana* leaf DNA, ranging from 5 to 500 ng per reaction (approximately $3.6\times10^4$ to $3.6\times10^6$ host *GI* template copies per reaction assuming a 135 Mb *A. thaliana* genome), were converted into hamPCR libraries. (**a**) 2% agarose gel of the prepared libraries. (**b**) Relative abundance (RA) of all amplicons in the reaction. The *A. thaliana GI* ASV appears to increase at the highest template concentrations, but remains of constant abundance through the standard template range of 5–100 ng. (**c**) The *A. thaliana GI* ASV has been removed and the bacterial ASVs have been rescaled to give bacterial relative abundance.

## Discussion 2: Synthetic template

The ITS1 region from *Agaricus bisporus* (3) was amplified with the forward (1) and reverse (2) primers below:

```
>1___ITS1_forward_for_ITS_G-46643
AGAAGGAGATATACCATGGcttggtcatttagaggaagtaa
>2___ITS-GI_reverse_for_ITS_G-46644
ACAGGTTGATTCGCCTCAATAgctgcgttcttcatcgatgc
>3___ITS1_Agaricus_bisporus
cttggtcatttagaggaagtaaAAGTCGTAACAAGGTTTCCGTAGGTGAACCTGCGGAAGGATCATTA
TTGAATTATGTTTTCTAGATGGGTTGTAGCTGGCTCTTCGGAGTATGTGCACGCCTGTCTGGACTTCA
TTTTCATCCACCTGTGCACCTTTTGTAGTCTTTTTCAGGTATTGGAGGAAGTGGTCAGCCTATCAGCTC
TTTGCTGGATGTAAGGACTTGCAGTGTGAAAACAGTGCTGTCCTTTACCTTGGCCATGGAATCTTTTTCC
TGTTAGAGTCTATGTTATTCATTATACTCTTAGAATGTCATTGAATGTCTTTACATGGGCTATGCCTA
TGAAAATTATTATACAACTTTCAGCAACGGATCTCTTGGCTCTCgcatcgatgaagaacgcagc
```

A portion of the *GIGANTEA* (*GI*) gene from *Arabidopsis thaliana* (3) was amplified with the forward (1) and reverse (2) primers below:

```
>1___ITS-GI forward for_GI_gene_G-46645
GCATCGATGAAGAACGCAGCtattgaggcgaatcaacctgt
>2___GI-16S_reverse_for_GI_gene_G-46646
CTGAGCCAGGATCAAACTCTcctgagccttcatctgaatgt
>Gigantea_partial_Arabidopsis
 tattgaggcgaatcaacctgtATCTAACAATCAAACTGCTAACCGTAAAAGTAGGAATGTCAAGGGA-
CAGGGACCTGTGGCAGCATTTGATTCATACGTTCTTGCTGCTGTTTGTGCTCTTGCCTGTGAGGTTCAGC
TGTATCCTATGATCTCTGGTCGGGGGAACTTTTCCAATTCTGCCGTGGCTGGAACTATTACAAAGCCTG
TAAAGATAAATGGGTCATCTAAAGAGTATGGAGCTGGGATTGACTCGGCAATTAGTCATACGCGCCGAA
TTTTGGCAATCCTAGAGGCACTCTTTTCATTAAAACCATCTTCTGTGGGGACTCCATGGAGTTACAGTTC
TAGTGAGATAGTTGCTGCGGCCATGGTTGCAGCTCATATTTCCGAACTGTTCAGACGTTCAAAGGCC
TTGACGCATGCATTGTCTGGGTTGATGAGATGTAAGTGGGATAAGGAAATTCATAAAGAGCATCATCA
TTATATAACCTCATAGATGTTCACAGCAAAGTTGTTGCCTCCATTGTTGACAAAGCTGAACCC
TTGGAAGCCTACCTTAAGAATACACCGGTTCAGAAGGATTCTGTGACCTGTTTAAACTGGAAACAAGA-
GAACACATGTGCAAGCACCACATGCTTTGATACAGCGGTGACATCCGCCTCAAGGACTGAAATGAA
TCCAAGAGGAAACCATAAGTATGCTAGacattcagatgaaggctcagg
```

A portion of the 16S rDNA from *Pseudomonas syringae pv. tomato* DC3000 (3) was amplified with the forward (1) and reverse (2) primers below:

```
>1___GI-16S_forward_for_16S_G-46647
ACATTCAGATGAAGGCTCAGGagagtttgatcctggctcag
>16S_ITSo_reverse_for_16S_G-46648
GTGGTAATGATCCTTCCGtaccttgttacgactt
>PstDC3000_16S_27 F-1492R

agagtttgatcctggctcagATTGAACGCTGGCGGCAGGCCTAACACATGCAAGTCGAGCGGCAGCACGGG
TACTTGTACCTGGTGGCGAGCGGCGGACGGGTGAGTAATGCCTAGGAATCTGCCTGGTAGTGGGGGA
TAACGCTCGGAAACGGACGCTAATACCGCATACGTCCTACGGGAGAAAGCAGGGGACCTTCGGGCC
TTGCGCTATCAGATGAGCCTAGGTCGGATTAGCTAGTTGGTGAGGTAATGGCTCACCAAGGCGACGATCCG
TAACTGGTCTGAGAGGATGATCAGTCACACTGGAACTGAGACACGGTCCAGACTCCTACGGGAGGCAGCAG
TGGGGAATATTGGACAATGGGCGAAAGCCTGATCCAGCCATGCCGCGTGTGTGAAGAAGGTCTTCGGATTG
TAAAGCACTTTAAGTTGGGAGGAAGGGCAGTTACCTAATACGTATCTGTTTTGACGTTACCGACAGAA
TAAGCACCGGCTAACTCTGTGCCAGCAGCCGCGGTAATACAGAGGGTGCAAGCGTTAATCGGAATTAC
TGGGCGTAAAGCGCGCGTAGGTGGTTTGTTAAGTTGAATGTGAAATCCCCGGGCTCAACCTGGGAACTGCA
TCCAAAACTGGCAAGCTAGAGTATGGTAGAGGGTGGTGGAATTTCCTGTGTAGCGGTGAAATGCGTAGATA
TAGGAAGGAACACCAGTGGCGAAGGCGACCACCTGGACTGATACTGACACTGAGGTGCGAAAGCG
TGGGGAGCAAACAGGATTAGATACCCTGGTAGTCCACGCCGTAAACGATGTCAACTAGCCGTTGGGAGCC
TTGAGCTCTTAGTGGCGCAGCTAACGCATTAAGTTGACCGCCTGGGGAGTACGGCCGCAAGGTTAAAAC
TCAAATGAATTGACGGGGGCCCGCACAAGCGGTGGAGCATGTGGTTTAATTCGAAGCAACGCGAAGAACC
TTACCAGGCCTTGACATCCAATGAATCCTTTAGAGATAGAGGAGTGCCTTCGGGAGCATTGAGACAGGTGC
TGCATGGCTGTCGTCAGCTCGTGTCGTGAGATGTTGGGTTAAGTCCCGTAACGAGCGCAACCCTTGTCC
TTAGTTACCAGCACGTTAAGGTGGGCACTCTAAGGAGACTGCCGGTGACAAACCGGAGGAAGGTGGGGA
TGACGTCAAGTCATCATGGCCCTTACGGCCTGGGCTACACACGTGCTACAATGGTCGGTACAGAGGG
TTGCCAAGCCGCGAGGTGGAGCTAATCTCACAAAACCGATCGTAGTCCGGATCGCAGTCTGCAACTCGAC
TGCGTGAAGTCGGAATCGCTAGTAATCGCGAATCAGAATGTCGCGGTGAATACGTTCCCGGGCCTTG
TACACACCGCCCGTCACACCATGGGAGTGGGTTGCACCAGAAGTAGCTAGTCTAACCTTCGGGGGGACGG
TTACCACGGTGTGATTCATGACTGGGGTGaagtcgtaacaaggta
```

The ITS region of *Hyaloperonospora arabidopsidis* (3) was amplified with the forward (1) and reverse (2) primers below:

```
>1___16S-ITSo_forward_for_ITSo_G-46649
AAGTCGTAACAAGGTAcggaaggatcattaccac
>2___ITSo_reverse_for_ITSo_G-46650
GTGGTGGTGGTGGTGCTCGAgagcctagacatccactgctg
>>3___ITS1_ H. arabidopsidis
```

```
cggaaggatcattaccaCACCTAAAAAACTTTCCACGTGAACCGTTTCAACCCAATAGTTGGGGGTCTTA
TTTGGCGGCGGCTGCTGGCTTAATTGTTGGCGGCTGCTGCTGAGTGAGCCCTATCAAAAAAAAGGCGAACG
TTTGGGCTTCGGCCTGATTTAGTAGTCTTTTTTTCTTTTAAACCCCTTCCTTAATACTGAATATACTG
TGGGGACGAAAGTCTCTGCTTTTAACTAGATAGCAACTTTcagcagtggatgtctaggct
```

The four amplicons were gel purified and combined by overlap extension PCR. First, the ITS amplicon was joined with the 16S rDNA amplicon, and the *GIGANTEA* amplicon was joined with the ITSo amplicon. These two fragments were gel purified and combined by overlap extension PCR to make the final fragment below, which was cloned into pGEM-T Easy (Promega, Madison, WI, USA).

```
>Synthetic_equimolar_templates agaaggagatataccatggcttggtcatttagaggaag-
taaAAGTCGTAACAAGGTTTCCGTAGGTGAACCTGCGGAAGGATCATTATTGAATTATGTTTTCTAGA
TGGGTTGTAGCTGGCTCTTCGGAGTATGTGCACGCCTGTCTGGACTTCATTTTCATCCACCTGTGCACC
TTTTGTAGTCTTTTTCAGGTATTGGAGGAAGTGGTCAGCCTATCAGCTCTTTGCTGGATGTAAGGAC
TTGCAGTGTGAAAACAGTGCTGTCCTTTACCTTGGCCATGGAATCTTTTTCCTGTTAGAGTCTATGTTA
TTCATTATACTCTTAGAATGTCATTGAATGTCTTTACATGGGCTATGCCTATGAAAATTATTATACAAC
TTTCAGCAACGGATCTCTTGGCTCTCgcatcgatgaagaacgcagctattgaggcgaatcaacctgtATC
TAACAATCAAACTGCTAACCGTAAAAGTAGGAATGTCAAGGGACAGGGACCTGTGGCAGCATTTGATTCA
TACGTTCTTGCTGCTGTTTGTGCTCTTGCCTGTGAGGTTCAGCTGTATCCTATGATCTCTGG
TCGGGGGAACTTTTCCAATTCTGCCGTGGCTGGAACTATTACAAAGCCTGTAAAGATAAATGGGTCATC
TAAAGAGTATGGAGCTGGGATTGACTCGGCAATTAGTCATACGCGCCGAATTTTGGCAATCCTAGAGGCAC
TCTTTTCATTAAAACCATCTTCTGTGGGGACTCCATGGAGTTACAGTTCTAGTGAGATAGTTGC
TGCGGCCATGGTTGCAGCTCATATTTCCGAACTGTTCAGACGTTCAAAGGCCTTGACGCATGCATTGTC
TGGGTTGATGAGATGTAAGTGGGATAAGGAAATTCATAAAAGAGCATCATCATTATATAACCTCATAGATG
TTCACAGCAAAGTTGTTGCCTCCATTGTTGACAAAGCTGAACCCTTGGAAGCCTACCTTAAGAA
TACACCGGTTCAGAAGGATTCTGTGACCTGTTTAAACTGGAAACAAGAGAACACATGTGCAAGCACCACA
TGCTTTGATACAGCGGTGACATCCGCCTCAAGGACTGAAATGAATCCAAGAGGAAACCATAAGTATGCTAG
acattcagatgaaggctcaggagagtttgatcctggctcagATTGAACGCTGGCGGCAGGCCTAACACA
TGCAAGTCGAGCGGCAGCACGGGTACTTGTACCTGGTGGCGAGCGGCGGACGGGTGAGTAATGCCTAGGAA
TCTGCCTGGTAGTGGGGGATAACGCTCGGAAACGGACGCTAATACCGCATACGTCCTACGGGAGAAAG-
CAGGGGACCTTCGGGCCTTGCGCTATCAGATGAGCCTAGGTCGGATTAGCTAGTTGGTGAGGTAATGGC
TCACCAAGGCGACGATCCGTAACTGGTCTGAGAGGATGATCAGTCACACTGGAACTGAGACACGGTCCA-
GACTCCTACGGGAGGCAGCAGTGGGGAATATTGGACAATGGGCGAAAGCCTGATCCAGCCATGCCGCGTG
TGTGAAGAAGGTCTTCGGATTGTAAAGCACTTTAAGTTGGGAGGAAGGGCAGTTACCTAATACGTATCTG
TTTTGACGTTACCGACAGAATAAGCACCGGCTAACTCTGTGCCAGCAGCCGCGGTAATACAGAGGG
TGCAAGCGTTAATCGGAATTACTGGGCGTAAAGCGCGCGTAGGTGGTTTGTTAAGTTGAATGTGAAA
TCCCCGGGCTCAACCTGGGAACTGCATCCAAAACTGGCAAGCTAGAGTATGGTAGAGGGTGGTGGAA
TTTCCTGTGTAGCGGTGAAATGCGTAGATATAGGAAGGAACACCAGTGGCGAAGGCGACCACCTGGACTGA
TACTGACACTGAGGTGCGAAAGCGTGGGGAGCAAACAGGATTAGATACCCTGGTAGTCCACGCCG
TAAACGATGTCAACTAGCCGTTGGGAGCCTTGAGCTCTTAGTGGCGCAGCTAACGCATTAAGTTGACCGCC
TGGGGAGTACGGCCGCAAGGTTAAAACTCAAATGAATTGACGGGGGCCCGCACAAGCGGTGGAGCATGTGG
TTTAATTCGAAGCAACGCGAAGAACCTTACCAGGCCTTGACATCCAATGAATCCTTTAGAGATAGAGGAG
TGCCTTCGGGAGCATTGAGACAGGTGCTGCATGGCTGTCGTCAGCTCGTGTCGTGAGATGTTGGGTTAAG
TCCCGTAACGAGCGCAACCCTTGTCCTTAGTTACCAGCACGTTAAGGTGGGCACTCTAAGGAGACTGCCGG
TGACAAACCGGAGGAAGGTGGGGATGACGTCAAGTCATCATGGCCCTTACGGCCTGGGCTACACACGTGC
TACAATGGTCGGTACAGAGGGTTGCCAAGCCGCGAGGTGGAGCTAATCTCACAAAACCGATCGTAG
TCCGGATCGCAGTCTGCAACTCGACTGCGTGAAGTCGGAATCGCTAGTAATCGCGAATCAGAATGTCGCGG
TGAATACGTTCCCGGGCCTTGTACACACCGCCCGTCACACCATGGGAGTGGGTTGCACCAGAAGTAGCTAG
TCTAACCTTCGGGGGGACGGTTACCACGGTGTGATTCATGACTGGGGTGaagtcgtaacaaggtacg-
gaaggatcattaccacACCTAAAAAACTTTCCACGTGAACCGTTTCAACCCAATAGTTGGGGGTCTTA
TTTGGCGGCGGCTGCTGGCTTAATTGTTGGCGGCTGCTGCTGAGTGAGCCCTATCAAAAAAAAGGCGAACG
TTTGGGCTTCGGCCTGATTTAGTAGTCTTTTTTTCTTTTAAACCCCTTCCTTAATACTGAATATACTG
TGGGGACGAAAGTCTCTGCTTTTAACTAGATAGCAAC
TTTcagcagtggatgtctaggcttcgagcaccaccaccaccac
```

## Discussion 3: Designing hamPCR HM-tagging primers

Investigators wishing to use hamPCR for a study system not demonstrated in this manuscript will need to design suitable HM-tagging primers specific to that system. Before embarking on new primer design, please first check our public resource of functional primers, which includes new

primer sequences and primer combinations contributed by the community: https://docs.google.com/spreadsheets/d/190VcSCMXmKEAIawp3GE-ZHXf1v1G3V7uuMTo4anYsJw/edit?

Here, we provide some helpful tips for new primer design. Researchers should have already decided on a two-step PCR system (see Appendix 1 - Discussion 1), which defines the universal overhangs that will be added to the HM-tagging primers.

## 1 Choose the microbial amplicon

For example, choose the V4 region of 16S rDNA. Calculate its approximate length (~285 bp) and GC content (~ 56%), and find the length and annealing/melting temperatures of the primers needed to amplify it (~ 73°C for 515F and ~ 52°C for 799R or 806R). Numerous melting temperature calculators are freely accessible online. At the time of writing, Thermo Fisher's 'Multiple Primer Analyzer' was our favorite.

## 2 Find candidate host genes

Research single or low-copy host genes in the host to be targeted. Such genes are often of interest for making phylogenies (*Li et al., 2017*) and therefore for many organisms have already been identified and written about. If sequences for candidate genes are publicly available, download candidate gene sequences into a text editor. If the host organism has no or few sequences available, look for candidate genes in two or more related organisms for which sequences are available. By aligning the genes in these related organisms, conserved regions in those genes can be identified that have a high chance of also being conserved in the host of interest. Those conserved areas may be used to design primers. Be wary of using the cDNA sequence of such genes, as the introns present in the genome are not part of the cDNA. Please note that this concern also applies to genes from eukaryotic microbes.

## 3 Find primer binding sites in the host candidate genes

Search the candidate genes for a region of similar length to the microbial amplicon that does not have long stretches of the same repeated base. Genes that appear by eye to have a random distribution of bases are excellent. If the candidate genes differ in GC content, start with the gene for which the GC content is most similar to the microbial amplicon. Locate a short (18–25 bp) region suitable for the forward primer that has a similar annealing temperature to that used for the microbial amplicon, erring on the side of a higher annealing temperature. Follow normal primer design guidelines for selecting a region (excellent advice and software are freely offered by many companies that sell oligos). When a candidate forward primer has been identified, look downstream in the host sequence for a similarly suitable reverse primer sequence. Ideally, the distance between the primers will differ from the length of the microbial amplicon by about 80–120 bp, allowing visualization of both amplicons on an agarose gel, and ensuring both amplicons will be sequenced with similar efficiency. This means for each forward primer candidate, there is about 40 bp of sequence in which to locate an ideal reverse primer candidate. A difference of fewer than 80 bp is also acceptable, especially if the host amplicon is not of overwhelming abundance and gel separation of the bands (as we show in *Figure 3*) is not necessary. Length and GC biases can be mitigated to some extent by choice of polymerase and adjusting cycling conditions (*Aird et al., 2011*; *Dabney and Meyer, 2012*).

## 4 Assemble the full HM-tagging primer sequences

Add the universal overhangs to each forward and reverse primer that will bind to the index primers. *Appendix 1—figure 6* below helps to visualize how the universal overhangs on the HM-tagging primers interact with the indexing PCR primers.

```
>PCR_F_G-40610
5' AATGATACGGCGACCACCGAGATCTACACTCTTTCCCTACACGACGCTCTTC
>Reverse_PCR_primer
5' CAAGCAGAAGACGGCATACGAGATNNNNNNNNNGTGACTGGAGTTCAGACGTGTGCTC
>515_F1_G-46603
5' TCCCTACACGACGCTCTTCCGATCT[GA]gtgycagcmgccgcggtaa 3
>799_R1_G-46601
5' GGAGTTCAGACGTGTGCTCTTCCGATCT[TG]cmgggtatctaatcckgtt 3'

OVERLAP ON SIDE OF FORWARD PRIMERS:
5' AATGATACGGCGACCACCGAGATCTACACTCTTTCCCTACACGACGCTCTTC 3'       <- PCR_F_G-40610
                515_F1_G-46603 -> 5' TCCCTACACGACGCTCTTCCGATCT[GA]gtg...
                                  3' AGGGATGTGCTGCGAGAAGGCTAGA[CT]cac...

OVERLAP ON SIDE OF REVERSE PRIMERS:
        ...[CA]AGATCGGAAGAGCACACGTCTGAACTCC 3'
        ...[GT]TCTAGCCTTCTCGTGTGCAGACTTGAGG 5' <- 799_R1_G-46601
Reverse_PCR_primer -> 3' CTCGTGTGCAGACTTGAGGTCAGTGNNNNNNNNNTAGAGCATACGGCAGAAGACGAAC 5'
```

**Appendix 1—figure 6.** Alignment of PCR primers with HM-tagged templates.

If there is sequence variation in the amplicon being targeted (which can be the case in particular for microbial amplicons such as ITS and 16S rDNA), a short linker sequence may be used, positioned just 5' of the template-binding part of the primer. The purpose of the linker sequence is to *not* bind to as many template sequences as possible. Thus, it helps the 5' end of the primer avoid contact with template molecules, which could bias amplification. Primers targeting host genes (for which sequence variation is not expected) will likely not benefit from such linker sequences. Keep in mind throughout that shorter HM-tagging primers tend to work better, so do not add unnecessary bases.

After deciding on candidate full primer sequences, align the full primer sequences to the host template again, and check if any of the 3' bases of the universal overhang also by chance match the host gene (this would effectively extend the template-specific portion of the primer). If any bases of the universal overhang can be considered as part of the template-specific portion of the primer, the 3' end of the template-specific portion of the primer may be shortened. For example, in the following primer that binds the wheat *PolA1* gene, the final 'ct' of the underlined universal overhang (highlighted purple) also is complementary to the *PolA1* template, and should be part of the length and annealing temperature calculation for the template-specific portion of the primer.

```
>Ta.PolA1_F1_G-46750
TCCCTACACGACGCTCTTCCGATct gatgttgtggaaggaattgaa
```

## 5 Check full primer sequences for predicted incompatibility in the same reaction

Some primers may have strong and unexpected complementarity to other primers or to themselves, leading them to form dimers rather than bind to the intended template. Dimerization can ruin the reaction. We used Thermo Fisher's 'Multiple Primer Analyzer', freely available online, to look for primer interaction. Many well-functioning combos will show some weak interactions - although it is important to minimize dimers, especially those with binding near the 3' end of the primer which polymerase could potentially extend into a product, it is not necessary to eliminate all predicted interactions. For example, primers for V4 16S rDNA and the *A. thaliana GI* gene (515_F1_G-46603, 799_R1_G-46601, At.GI_F1_G-46602, and At.GI_R502bp_G-46614) are very robust together, and they show potential weak dimers as follows:

```
Self-Dimers:
2 dimers for: At.GI_F1_G-46602
5-tccctacacgacgctcttccgatctgtaaagataaatgggtcatctaa->
          ||  |  |      ||||  ||  ||||     |   |  ||
      <-aatctactgggtaaatagaaatgtctagccttctcgcagcacatccct-5
     5-tccctacacgacgctcttccgatctgtaaagataaatgggtcatctaa->
          ||     ||  |  |  ||||  |  ||      ||
```

```
<-aatctactgggtaaatagaaatgtctagccttctcgcagcacatccct-5
Cross Primer Dimers:
515_F1_G-46603 with At.GI_F1_G-46602
515_F1_G-46603
  5-tccctacacgacgctcttccgatctgtaaagataaatgggtcatctaa->
     ||        |   |     || |||| | | ||      ||
<-aatggcgccgmcgacygtgagtctagccttctcgcagcacatccct-5
```

## 6 Experimentally test primer candidates

Primers sometimes defy predictions, working better or worse than expected. For example, if primers for which dimerization is not predicted nonetheless have some off-target complementarity to microbial or host genomes, they may not provide clear products. We recommend ordering at least three host primer pairs at once, and testing all of them. First, test each primer pair alone in a normal (not hamPCR) reaction of 35 cycles using a sample containing host DNA as template, and ensure that a product of the correct size is formed. Primers that work well alone can then be tested in combination with microbial primers in hamPCR. For initial testing, it is helpful to a DNA template that has an equal amount of host and microbial templates, so that both bands are expected. For example, host and microbial templates can be mixed in an equimolar ratio (which takes into account any differences in genome size between host and the microbial target). Alternatively, template amplicons can be cloned to make a single chimeric DNA fragment as we described in Appendix 1 - Discussion 2, forming a template with a known 1:1 host-to-microbe ratio.

