## [Decision Letter]

Thank you for submitting your article "Host-associated microbe PCR (hamPCR) enables convenient measurement of both microbial load and community composition" for consideration by *eLife*. Your article has been reviewed by 3 peer reviewers, and the evaluation has been overseen by Becky Bart as the Reviewing Editor and Meredith Schuman as the Senior Editor. The following individuals involved in review of your submission have agreed to reveal their identity: Mingsheng Qi (Reviewer #1); Benjamin E. Wolfe (Reviewer #2).

Essential revisions:

1. One thing that may prevent wide adoption of this technique is if significant optimization is required. We appreciate that the authors provided some details in the Supplement and these helpful tips may make it easier for others to develop hamPCR for their own systems. However, given that this is a methods paper and impact will be determined by adoption, we think this aspect of the paper should be expanded further. For example, appropriate selection of host genes seems like a particularly important step in this approach. Please include additional strategy discussion on this topic as well as any nuance for primer design with the overall goal of making it easier for this method to be applied to diverse systems.

2. The approach seems to struggle at the high and low ends of host:microbe ratios. For example, with high levels of microbial infection, the authors note that hamPCR has reduced accuracy. The authors propose three solutions to this problem (1. altering host:microbe amplicon ratio, 2. use a host gene with higher copy number, 3. adjust concentrations of host primers). Data is presented for #1 and #3. Is there any data to support that #2 would actually work? The outlier in Figure 5B also caught our eyes as potentially worrisome. Is it possible to repeat the hamPCR for this outlier to confirm the striking difference from the other three samples in the eds1-1 Hpa + Pto sample?

3. Figure 1d. y axis title, should be Load (M:H ratio) 4th-root axis.

4. Could the DNA extraction method used cause biases in hamPCR for/against either the host or the microbiome? If two different labs study the same system (let's say bacterial communities growing on Arabidopsis leaves) but use different DNA extraction approaches, would we expect them to obtain different answers using hamPCR? Did the authors try several different DNA extraction methods to see if this is an issue? Or has another team of researchers considered this and addressed it in a separate paper? I would appreciate seeing either data to address this or a discussion paragraph that reasons through this.

5. Another technical aspect, can the authors provide evidence that, even for rare OTU counts, there will not be loss during gel isolation? If not, this claim should be softened.

6. One emerging theme in microbiome science is to have consistent methodologies that are used across studies/labs to allow direct comparisons of microbiome datasets. Standardization of approaches may make microbiome science more robust in the long-term. Given much of the nuance in developing hamPCR for different systems, my impression is that this method is best for comparing samples within a particular host-microbe system and not across systems. For example, it may be challenging to directly compare my bacterial load hamPCR data from Arabidopsis to another lab's if we used different Arabidopsis host genes or if we used different 16S gene regions. Can the authors unpack this a bit in a discussion paragraph? If this approach is widely adopted, is there a way to standardized hamPCR so that it can be consistently used and compared across datasets?

7. There appears to be considerable non-specific amplification or dimers in the gels presented throughout the manuscript. Could this non-specific amplification vary across host-microbe primer combinations? Would this impact quantification of host and microbial amplicons?

8. Line 466: "true" may be an inappropriate word here. It implies that you have discovered the absolute truth about correlations between bacteria in this system using your new approach. This claim would need to be supported with direct experimentation and manipulation so should be softened in this paper.

9. The authors nicely summarize many of the inherent technical problems for a number of techniques, but overlook discussion or investigation of the biases introduced when PNAs are part of the library preparation. PNAs do not uniformly block host in different host systems and can inhibit the capture of certain microbial taxa in amplicon sequencing surveys. Did the authors perform any sequencing to generate a reference sample without PNA? Would this impact the ability to reconstruct the relative abundances?

10. Part of inspiring readers to adopt this approach is to clearly paint the biologically-driven potential in the narrative. We ask the authors to look for places in the text where this emphasis can be strengthened. For example, beyond their status as crops with different genome size, is there additional rationale for the selection of wheat, pepper, and corn in Figures 5 and 6? Likewise, while it is very cool that hamPCR also works in P. pacificus is there more logic behind its selection as a host system?

*Reviewer #2 (Recommendations for the authors):*

Figure 1d y-axis label appears to have a typo (4th-4oot Axis should be 4th-Root Axis)

Line 466: "true" may be an inappropriate word here. It implies that you have discovered the absolute truth about correlations between bacteria in this system using your new approach. I think it would take direct experimentation and manipulation to have that level of confidence.

I was confused with the use of the word "aligned" in Figure 6 and related text.

It might be helpful in Figure 1a and in all other instances to make 'tagging' more specific and call it 'source tagging' or 'host/microbe tagging.' This is not necessary, but might add some clarity to the terms used throughout the manuscript.

I know that the figure legends are already pretty long, but some additional details could help readers have additional clarity. For example, in Figure 5a and 5b, what do each of the columns within a treatment represent?

*Reviewer #3 (Recommendations for the authors):*

Throughout the manuscript, it would be helpful if more effort was paid to providing the reader with more context regarding the importance of each set of experiments and why each system was chosen.

– While the authors show the accuracy of the remixing the describe in Figure 3, the importance of being able to do isn't clearly concluded.

– The authors nicely summarize many of the inherent technical problems for a number of techniques, but overlook discussion or investigation of the biases introduced when PNAs are part of the library preparation. PNAs do not uniformly block host in different host system and can inhibit the capture are certain microbial taxa in amplicon sequencing surveys. Did the authors perform any sequencing to generate a reference samples without PNA? Would this impact the ability to reconstruct the relative abundances?

– The authors note in their discussion of Figure 4 findings that the primer set that amplifies V5V6V7 predicts a different load and community composition (e.g., lower Sphingobacteriaceae and higher Oxalobacteriaceae). Have such differences in captured taxa match with primer bias demonstrated in previous studies?

– Beyond their status as crops with different genome size, is there more rationale for the selection of wheat, pepper, and corn in Figures 5 and 6? Likewise, while its very cool that hamPCR also work in P. pacificus is there more logic behind its selection as a host system?

---

## [Author Response]

Essential revisions:1. One thing that may prevent wide adoption of this technique is if significant optimization is required. We appreciate that the authors provided some details in the Supplement and these helpful tips may make it easier for others to develop hamPCR for their own systems. However, given that this is a methods paper and impact will be determined by adoption, we think this aspect of the paper should be expanded further. For example, appropriate selection of host genes seems like a particularly important step in this approach. Please include additional strategy discussion on this topic as well as any nuance for primer design with the overall goal of making it easier for this method to be applied to diverse systems.

A deeper discussion about the design of suitable host primers has been added to the Supplementary Information as Supplementary Discussion 3, and is now mentioned in the main text in the first section of the Methods.

2. The approach seems to struggle at the high and low ends of host:microbe ratios. For example, with high levels of microbial infection, the authors note that hamPCR has reduced accuracy. The authors propose three solutions to this problem (1. altering host:microbe amplicon ratio, 2. use a host gene with higher copy number, 3. adjust concentrations of host primers). Data is presented for #1 and #3. Is there any data to support that #2 would actually work?

Both qPCR and amplicon sequencing can be used to detect copy number variation in genomes [1]. Because amplicon-based methods are known to be sensitive to small differences in gene copy number, we are confident, without generating additional data on the topic, that #2 would work.

Furthermore, bacterial genomes from different taxa are known to vary slightly in their copy number of 16S rDNA, usually from between 1 to about 15 copies [2]. These variations are reflected in sequence counts from amplicon sequencing, biasing the counts towards taxa with more 16S rDNA gene copies [2, 3, 4]. This phenomenon has been well documented, distorts the accurate description of microbial communities, and therefore has led to some efforts to correct 16S rDNA gene amplicon data by dividing the counts from each taxon by the (estimated) 16S rDNA copy number of that taxon, so that the counts better reflect the numbers of bacterial cells.

Because amplicon methods are sensitive to copy number variation (whether those copies are from inside the same cell, or coming from different cells), we reasoned that choosing a host gene with a higher copy number, similar to the effects of copy number variation on 16S rDNA gene counts, will increase the representation of that host amplicon in the final library (because there will be more template host DNA molecules available to amplify). We did not test this explicitly – we think the evidence from literature is strong support on its own. We have added to the paper a statement that now references the Kembel 2012 paper, which we hope adequately supports our claim:

“Second, a host gene with a higher copy number could be chosen for HM-tagging throughout the entire project, which would increase host representation by a factor of that copy number (Kembel et al., 2012).”

(1) Martins, W.F.S., Subramaniam, K., Steen, K. et al. Detection and quantitation of copy number variation in the voltage-gated sodium channel gene of the mosquito *Culex quinquefasciatus*. Sci Rep 7, 5821 (2017). https://doi.org/10.1038/s41598-017-06080-8

(2) Kembel, S. W., Wu, M., Eisen, J. A., and Green, J. L. (2012). Incorporating 16S gene copy number information improves estimates of microbial diversity and abundance. PLoS Computational Biology, 8(10), e1002743. https://doi.org/10.1371/journal.pcbi.1002743

(3) Starke, R., Pylro, V. S., and Morais, D. K. (2021). 16S rRNA Gene Copy Number Normalization Does Not Provide More Reliable Conclusions in Metataxonomic Surveys. Microbial Ecology, 81(2), 535–539. https://doi.org/10.1007/s00248-020-01586-7

(4) Louca, S., Doebeli, M., and Parfrey, L. W. (2018). Correcting for 16S rRNA gene copy numbers in microbiome surveys remains an unsolved problem. Microbiome, 6(1), 41. https://doi.org/10.1186/s40168-018-0420-9

The outlier in Figure 5B also caught our eyes as potentially worrisome. Is it possible to repeat the hamPCR for this outlier to confirm the striking difference from the other three samples in the eds1-1 Hpa + Pto sample?

Unfortunately the DNA for this sample has been used up and it cannot be resequenced. However, we agree that this sample is quite curious and would benefit from more explanation. Looking more closely at the data from this sample, we saw that the CFU counts for *Pst* DC3000 in this sample were also 1.5 times higher than the next highest sample (this sample is also visible as an outlier in the CFU counts in panel 5c). Because this independent metric showed this sample to be an outlier for pathogen load, we suspect that this sample was actually an outlier for biological reasons (or for technical reasons unrelated to the hamPCR technique itself, such as the possibility of a failure to remove a bacterial-laden soil particle from the leaves of this seedling prior to DNA extraction). This information has been added to the text (see below).

Although the host amplicon represented only 1.16% of the total sequence counts for this sample (which was under our recommendation of a host amplicon abundance of at least 3%), we noticed the depth of total sequencing reads in this sample was actually very high (410,326 reads), with 4,782 reads assigned to the host, and we began to question our 3% threshold. We realized our 3% threshold as defined in Supplementary Figure 4 was too simplistic, because it assumed a constant sequencing depth of approximately 10,000 reads. However, as sequencing depth increases, the sensitivity of the microbial load quotient to sampling noise (such as adding or subtracting a sequence count) decreases (eg. the difference between the quotients of 100,000/1,000 vs. 99,999/1,001 is much smaller than the difference between the quotients of 100/1 vs. 99/2). That is to say, with deeper sequencing, lower percentages of host amplicons can be tolerated without increasing noise in the microbe/host ratio.

We therefore calculated a new metric, a noise level *N*, that takes into account both the abundance of the host amplicon and the sequencing depth, illustrated and explained in a new panel (d) in Supplementary Figure 4. Samples with *N* greater than 0.22 (corresponds to a host abundance of 3% at 10,000 reads) we consider noisy, and samples with *N* less than or equal to 0.22 we consider acceptable. Of course, the acceptable level of *N* can be set by the researcher. By this improved and more logical threshold, the outlier in Figure 5b falls well within our reliability threshold. We believe the most likely explanation for this outlier is indeed unrelated to hamPCR. Therefore we still point out the outlier in Figure 5b, but further also point out the corresponding outlier in the CFU counts, and discuss this analysis briefly in the manuscript.

“We noted one outlier sample (red arrows, Figure 5b, 5c) with especially high microbial load. This sample was also an outlier by CFU counting, and because hamPCR fell within the quantitative range defined by host abundance and sequencing depth (Supplementary Figure 4), we conclude that this outlier was likely not due to limitations of hamPCR.”

We also reanalyzed the pepper growth curve in Figure 5 that had samples with low host abundances using this new noise threshold *N*. Indeed, the majority of samples in our grey-boxed area were above the acceptable noise limits. We edited the text there as well to remove references to a simple 3% threshold and refer the reader to Supplementary Figure 4d for details on noise thresholds:

“By 7 dpi, bacterial growth had reduced the host *GI* amplicon abundances for most samples to levels at which load calculations become less reliable at their sequencing depth (as defined in Supplementary Figure 4d); this was also the case at 9 and 11 dpi (gray box, Figure 6a).”

3. Figure 1d. y axis title, should be Load (M:H ratio) 4th-root axis

Indeed, thank you.

4. Could the DNA extraction method used cause biases in hamPCR for/against either the host or the microbiome? If two different labs study the same system (let's say bacterial communities growing on Arabidopsis leaves) but use different DNA extraction approaches, would we expect them to obtain different answers using hamPCR? Did the authors try several different DNA extraction methods to see if this is an issue? Or has another team of researchers considered this and addressed it in a separate paper? I would appreciate seeing either data to address this or a discussion paragraph that reasons through this.

Differences in DNA extraction method will certainly change the results, not only of the microbe-to-plant ratio, but also in the representation of microbes, because microbes differ in their sensitivity to different lysis methods. This is a well-documented concern in microbiome studies and has been demonstrated by using different methods on the same mock community in papers such as the following:

Yuan, S., Cohen, D. B., Ravel, J., Abdo, Z., and Forney, L. J. (2012). Evaluation of methods for the extraction and purification of DNA from the human microbiome. PloS One, 7(3), e33865. https://doi.org/10.1371/journal.pone.0033865

Albertsen, M., Karst, S. M., Ziegler, A. S., Kirkegaard, R. H., and Nielsen, P. H. (2015). Back to Basics--The Influence of DNA Extraction and Primer Choice on Phylogenetic Analysis of Activated Sludge Communities. PloS One, 10(7), e0132783. https://doi.org/10.1371/journal.pone.0132783

In short, if the DNA is not extracted because plant or microbial cells are not lysed, it cannot be amplified in PCR. However, there is a good overall strategy to minimize the problem, as also proposed in the above papers, and that is to err on the side of a harsher lysis (using strong bead beating, as we have done), since this will leave fewer cells unlysed (and thus less information will be hidden). We note that similar concerns about lysis methods changing results also apply to DNA extraction for qPCR and live bacterial isolation for CFU counting (for which too harsh a lysis will kill bacteria, but too gentle a lysis will not release them from host tissue).

We addressed this in two places. First, in the Results section we mention briefly the following:

“All DNA preps employed heavy bead beating to ensure thorough lysis of both host and microbes, as an incomplete DNA extraction can lead to underrepresentation of hard-to-lyse cells (Albertsen et al., 2015; Yuan et al., 2012).”

Second, we added a paragraph to the discussion about sample selection and DNA extraction as follows:

“Because hamPCR can only quantify the DNA available in the template, choice of sample and appropriate DNA extraction methods are very important. […] Especially for short reads, as we have used here, this fragmentation is not a problem, and we recommend to err on the side of a harsher lysis, using strong bead beating potentially preceded by grinding steps using a mortar and pestle as necessary for tougher tissue.”

5. Another technical aspect, can the authors provide evidence that, even for rare OTU counts, there will not be loss during gel isolation? If not, this claim should be softened.

Of course, a gel extraction will not purify any DNA fragments that are outside the area of the section that has been cut from the gel. This would be a concern for some ITS amplicons, because these can vary in length and some may occur outside of the area most clearly visible on the gel as a “band”. However, for 16S rDNA amplicons, the total sequence length variation between different taxa is low, and one is unlikely to bias against certain taxa by cutting out the band with a small safety margin.

Another potential aspect of the reviewer’s concern relates to efficiency of gel extraction. Gel extraction certainly is not 100% efficient, and therefore a random subset of the molecules is lost. *Extremely* rare sequences would therefore be totally lost. However, such sequences are not of practical concern. This is because the absolute number of molecules that we successfully purify from the gel is in the order of many billions (often we measure several nanograms of DNA after gel extraction). This is far more than one would ever sequence from a single amplicon sample, so prior to sequencing the library must anyway be further diluted (thus further eliminating additional ultra-rare sequences).

When gel isolation (the technique shown in Figure 3) is used to increase the fraction of microbial reads and decrease the fraction of host reads in the dataset, this process will actually *increase* the ability to detect rare microbial sequences because it effectively allows deeper sampling of microbial populations, with fewer rare sequences diluted out prior to sequencing.

We also would like to point out that gel extraction of the 16S rDNA band has been used in many influential publications [e.g., 1, 2] using the 16S rDNA primers 799F and 1192R, which target the V5V6V7 region. As we show in Supplementary Figure 3c, these primers produce a large mitochondrial band of about 900 bp, which we and other researchers remove by gel-purifying the 16S rDNA band that migrates at about 500 bp. Although widespread use of gel purification is not evidence that the technique is without any issues, the literature does suggest that the advantages outweigh any disadvantages.

1) Bulgarelli, D., Rott, M., Schlaeppi, K. et al. Revealing structure and assembly cues for Arabidopsis root-inhabiting bacterial microbiota. Nature 488, 91–95 (2012). https://doi.org/10.1038/nature11336

2) Durán, P., Thiergart, T., Garrido-Oter, R., Agler, M., Kemen, E., Schulze-Lefert, P., and Hacquard, S. (2018). Microbial Interkingdom Interactions in Roots Promote Arabidopsis Survival. Cell, 175(4), 973–983.e14. https://doi.org/10.1016/j.cell.2018.10.020

6. One emerging theme in microbiome science is to have consistent methodologies that are used across studies/labs to allow direct comparisons of microbiome datasets. Standardization of approaches may make microbiome science more robust in the long-term. Given much of the nuance in developing hamPCR for different systems, my impression is that this method is best for comparing samples within a particular host-microbe system and not across systems. For example, it may be challenging to directly compare my bacterial load hamPCR data from Arabidopsis to another lab's if we used different Arabidopsis host genes or if we used different 16S gene regions. Can the authors unpack this a bit in a discussion paragraph? If this approach is widely adopted, is there a way to standardized hamPCR so that it can be consistently used and compared across datasets?

We have added the following paragraph to the discussion:

“hamPCR is best suited to comparing microbial loads within an experimental system using consistent host and microbe primers, because different microbe primer pairs can differ in their amplification efficiency and therefore yield different load measurements on the same DNA (Figure 4); using V5V6V7 primers to study *A. thaliana* will give different absolute load measurements than using another popular primer set, for the V4 region. […] By using hamPCR to sequence standards along with experimental samples, the measured microbial loads can be correlated with known “standardized” load ratios, which in turn can be compared across primer sets and labs.”

In addition, we have prepared a community-editable primer resource. This is a Google Document that curates known working combinations of primers. It includes all the working combinations from the paper. It includes mixes of the HM-tagging primers used in the publication, and will include additional working combinations. The community can contribute by supplying their own information as “comments” on the publicly accessible document. We will briefly check the contributions for completeness and appropriateness, and then post them along with the contributor’s name. We have added a reference to the community resource in Supplementary Table 1 where primer sequences are provided, and also in Supplementary Discussion 3 which was added to discuss primer design considerations.

https://docs.google.com/spreadsheets/d/190VcSCMXmKEAIawp3GE-ZHXf1v1G3V7uuMTo4anYsJw/edit?usp=sharing

7. There appears to be considerable non-specific amplification or dimers in the gels presented throughout the manuscript. Could this non-specific amplification vary across host-microbe primer combinations? Would this impact quantification of host and microbial amplicons?

Non-specific amplification / dimers do vary across host-microbe primer combinations. Indeed, they also vary between common 16S rRNA primer pairs used on their own (not shown). Fortunately non-specific amplicons amplified during the exponential PCR step do not, at least with our method, seem to impact quantification of host and microbial amplicons.

One reason is that non-specific amplicons can be recognized by their sequence and ignored. After the sequences of the amplicons have been extracted from the short read data, only those that match expected length and sequence patterns of the targeted amplicons need to be counted. Non-specific amplicons are certainly a nuisance because they represent wasted sequencing resources, but they can be excluded bioinformatically and therefore do not change the accuracy of the microbial load measurement. This is in contrast to ddPCR/qPCR, for which any off-target amplicons are also quantified!

A second reason is that the sensitive exponential amplicon step of hamPCR is done with a single primer pair. Off-target sequences do squander PCR reagents including primers and dNTPs, such that they become limiting at earlier cycles than without off-target sequences, but because the exponential PCR step is done with a single primer pair, such inferior amplification conditions are shared by all molecules, and therefore do not differentially affect the host or microbial amplicon. Any off-target binding occurring in the initial tagging reaction (before the PCR step) would certainly be a concern if the reaction was carried on long enough, because for example the microbial primer pair might become limiting at an earlier cycle number, leading to underestimates of microbial load. However, limiting the tagging cycle to a low number of cycles ensures that – should primers targeting a particular host or microbial amplicon be non-specific – the fraction still available to bind the correct sequence remains in excess.

8. Line 466: "true" may be an inappropriate word here. It implies that you have discovered the absolute truth about correlations between bacteria in this system using your new approach. This claim would need to be supported with direct experimentation and manipulation so should be softened in this paper.

We deleted the word “true”, as the sentence makes our intended point without this word.

9. The authors nicely summarize many of the inherent technical problems for a number of techniques, but overlook discussion or investigation of the biases introduced when PNAs are part of the library preparation. PNAs do not uniformly block host in different host systems and can inhibit the capture of certain microbial taxa in amplicon sequencing surveys. Did the authors perform any sequencing to generate a reference sample without PNA? Would this impact the ability to reconstruct the relative abundances?

Although the peptide nucleic acids (PNAs) we used for the plant samples are extremely useful for blocking chloroplast and mitochondria in the majority of plant hosts, and produce very minimal bias, it is correct that there are some limitations regarding unintentionally blocked bacterial taxa, further investigated here:

Jackrel, S. L., Owens, S. M., Gilbert, J. A., and Pfister, C. A. (2017). Identifying the plant-associated microbiome across aquatic and terrestrial environments: the effects of amplification method on taxa discovery. Molecular Ecology Resources, 17(5), 931–942. https://doi.org/10.1111/1755-0998.12645

Also, there are some limitations regarding their utility blocking organellar DNA from some plant hosts, further investigated here:

Fitzpatrick, C. R., Lu-Irving, P., Copeland, J., Guttman, D. S., Wang, P. W., Baltrus, D. A., Dlugosch, K. M., and Johnson, M. T. J. (2018). Chloroplast sequence variation and the efficacy of peptide nucleic acids for blocking host amplification in plant microbiome studies. Microbiome, 6(1), 144. https://doi.org/10.1186/s40168-018-0534-0

hamPCR users may also opt for other strategies to overcome excess organellar DNA. For example, the V5V6V7 16S rDNA primers themselves exclude amplification of chloroplast sequences and amplify mitochondria as a band of a different size that can be excised by gel extraction (Supplementary Figure 3c). We did not use PNA when employing the V5V6V7 primers. If the organellar sequences in a given host are not overwhelming, it may not be necessary to employ any strategy to reduce off-target organellar amplification.

Regardless of whether or not amplification of the majority of organellar templates is blocked, any remaining organellar amplicons that go on to be sequenced can be removed from the data bioinformatically. We expect that, after bioinformatic curation, the microbial load of a sample for which PNA was employed will be the same as for a sample for which PNA was not employed. However, we did not test this explicitly. Because PNA is a potential source of variability, we recommend being consistent with it across all samples to be compared (i.e. PNA should either be used for all samples in the dataset or none).

We have added to the text at our first mention of PNAs:

“We note that although these PNAs are widely used and extremely effective, they do not work for all plant hosts (Fitzpatrick et al., 2018) and they can interfere with the analysis of certain bacteria in some environments (Jackrel et al., 2017).”

10. Part of inspiring readers to adopt this approach is to clearly paint the biologically-driven potential in the narrative. We ask the authors to look for places in the text where this emphasis can be strengthened. For example, beyond their status as crops with different genome size, is there additional rationale for the selection of wheat, pepper, and corn in Figures 5 and 6? Likewise, while it is very cool that hamPCR also works in P. pacificus is there more logic behind its selection as a host system?

We have edited the text throughout with attention to this comment, trying to strengthen the biological potential and offer more rationale.

We explain that *P. pacificus*, as a nematode, represents a whole group of small organisms on which hamPCR is useful. As the only non-plant in our paper, it plays a key role in helping to illustrate the broader potential of this technique. We allude to this and also reveal our choice of this particular nematode was convenience-driven:

“To demonstrate the utility of hamPCR outside of plants, we prepared samples of the nematode worm *Pristionchus pacificus*. Small hosts like nematodes where the entire organism is typically lysed during DNA preparation are ideal for hamPCR; the unique choice of this nematode was due to the convenience of a lab specializing in studies of this species in our institute (Sommer, 2006).”

To our justification of wheat, we add a note that it is hexaploid, demonstrating that hamPCR works with host genes that are not single-copy.

“We similarly validated the technique for fungal and bacterial microbes of *Triticum aestivum* (bread wheat), because it is the most widely grown crop in the world yet one of the most difficult to study due to being hexaploid and having a 16 Gb haploid genome.”

We also added the word “polyploid” to the legend title of figure 5.

Figure 5. hamPCR can be generalized to more than two amplicons, non-plant hosts, and large/polyploid host genomes.

We added to our motivating statement for our demonstrations with pepper and corn.

**“**To go beyond model organisms and controlled titrations and to demonstrate the ability of hamPCR to yield biological insights into complex study systems, we conducted two experiments with crop plants.”

Reviewer #2 (Recommendations for the authors):Figure 1d y-axis label appears to have a typo (4th-4oot Axis should be 4th-Root Axis)

This was addressed as point number 3 under “essential revisions”.

Line 466: "true" may be an inappropriate word here. It implies that you have discovered the absolute truth about correlations between bacteria in this system using your new approach. I think it would take direct experimentation and manipulation to have that level of confidence.

This was addressed as point number 8 under “essential revisions”.

I was confused with the use of the word "aligned" in Figure 6 and related text.

Since what we mean by the “alignment” is actually mathematical scaling, we have replaced aligned with “scaled” and added other slight clarifications in the text. We hope this is clearer now.

It might be helpful in Figure 1a and in all other instances to make 'tagging' more specific and call it 'source tagging' or 'host/microbe tagging.' This is not necessary, but might add some clarity to the terms used throughout the manuscript.

We agree that “tagging” can be misunderstood, as it is also sometimes used for the general process of adding index sequences, and could therefore be confusing.

To avoid this confusion, we define a new term, “HM-tagging”, at the first mention of tagging as follows: “The first 2-cycle host-and-microbe template tagging step (“HM-tagging”) of…”

We think that adding a qualifying word such as “source tagging” or “host microbe tagging” throughout each usage in the text would complicate the flow reading in some sections, but we hope the new single-word term HM-tagging is a suitable compromise.

I know that the figure legends are already pretty long, but some additional details could help readers have additional clarity. For example, in Figure 5a and 5b, what do each of the columns within a treatment represent?

We have added the following statement to Figure 5a: “Each column represents an independent plant.”, as well as some other clarifying language.

Reviewer #3 (Recommendations for the authors):Throughout the manuscript, it would be helpful if more effort was paid to providing the reader with more context regarding the importance of each set of experiments and why each system was chosen. While the authors show the accuracy of the remixing the describe in Figure 3, the importance of being able to do isn't clearly concluded.

This is somewhat related to essential revision #10. In addition to changes mentioned there, we have now introduced the section that uses hamPCR with host, 16S rDNA, ad oomycete amplicons as follows:

“One disadvantage of 16S rDNA primers is that they readily amplify sequences from bacterial species, but their targets are absent from other microbes such as fungi, oomycetes, and archaea; other primer sets must therefore be used to detect these other groups. We attempted to overcome this limitation by setting up hamPCR not only with 16S rDNA and host *GI* sequences as described above, but also adding a third primer pair targeting oomycetes, which include important pathogens of *A. thaliana*.”

With regards to the remixing in Figure 3, we have added some more language explaining our motivation. We now introduce that section of the results:

“Some host DNA must be present so that microbial load can be calculated, but sequencing too much host DNA would add unnecessary expense.”

And we conclude that section of the results:

“Thus, if hamPCR libraries have already been prepared and the host amplicon is overabundant, the host representation in the pooled libraries can be easily and accurately reduced using basic agarose gel technologies to enable more efficient sequencing.”

– The authors nicely summarize many of the inherent technical problems for a number of techniques, but overlook discussion or investigation of the biases introduced when PNAs are part of the library preparation. PNAs do not uniformly block host in different host system and can inhibit the capture are certain microbial taxa in amplicon sequencing surveys. Did the authors perform any sequencing to generate a reference samples without PNA? Would this impact the ability to reconstruct the relative abundances?

This was addressed as point number 9 under “essential revisions”.

– The authors note in their discussion of Figure 4 findings that the primer set that amplifies V5V6V7 predicts a different load and community composition (e.g., lower Sphingobacteriaceae and higher Oxalobacteriaceae). Have such differences in captured taxa match with primer bias demonstrated in previous studies?

Indeed, biases between these or similar primer pairs have been studied in several publications. Slight variations in both amplification efficiency and classification accuracy of different taxa have been observed for different primer pairs. In the following study, a bias of V5V6V7 primers towards amplification of the class betaproteobacteria (which includes Oxalobacteriaceae) is noted:

Thijs, S., Op De Beeck, M., Beckers, B., Truyens, S., Stevens, V., Van Hamme, J. D., Weyens, N., and Vangronsveld, J. (2017). Comparative Evaluation of Four Bacteria-Specific Primer Pairs for 16S rRNA Gene Surveys. Frontiers in Microbiology, 8, 494. https://doi.org/10.3389/fmicb.2017.00494

Regarding the reduction of Sphingobacteriaceae in the V5V6V7 dataset – this may be influenced by mismatches between the binding site of the 1192R primer and many Sphingobacteriaceae sequences:

1192R primer: ACGTCATCCCCACCTTCCReverse complement of 1192R: GGAAGGTGGGGATGACGTGammaproteobacteria1: GGAAGGTGGGGATGACGTGammaproteobacteria2: GGAAGGTGGGGATGACGTFlavobacteriaceae: GGAAGGTGGGGATGACGTSphingobacteriaceae1: GGAAGGAGGGGACGACGTSphingobacteriaceae2: GGAAGGTGGGGACGACGTSphingobacteriaceae3: GGAAGGAGGGGACGACGTSphingobacteriaceae4: GGAAGGAGGGGACGACGT* * * * * * * * * * * * * * * *

For these Sphingobacteriaceae sequences, such mismatches are not present at the binding sites for the other primer sets.

The amplification biases in 16S rDNA that we see between our hamPCR datasets are reasonably explained by features of the 16S rDNA primers themselves. This, combined with the evidence from Figure 2 that for V4 16S rDNA primers, the microbial community is the same whether or not the host amplicon is included and co-amplified, lets us conclude that taxonomic differences in microbial community composition are not caused by the hamPCR protocol.

We added a citation of the above paper to the manuscript as follows:

“…with slight deviations likely due to the different taxonomic classification pipeline used for the metagenome reads (Regalado et al., 2020), as well as known biases resulting from amplification or classification of different 16S rDNA variable regions (Graspeuntner et al., 2018; Thijs et al., 2017).”

– Beyond their status as crops with different genome size, is there more rationale for the selection of wheat, pepper, and corn in Figures 5 and 6? Likewise, while its very cool that hamPCR also work in P. pacificus is there more logic behind its selection as a host system?

[This was addressed as point number 10 under “essential revisions”].